



# 1 Hydrological regime index for non-perennial rivers

Pablo F. Dornes[1], Rocío N. Comas[1]
[1] Facultad de Ciencias Exactas y Naturales, Universidad Nacional de La Pampa (UNLPam), Av. Uruguay 151 (6300), Santa
Rosa, La Pampa, Argentina
*Correspondence to*: Pablo F. Dornes (pablodornes@exactas.unlpam.edu.ar)
**Abstract.** The hydrological regime is an integrated basin response that constitutes an established paradigm for environmental
flows (E-Flows) to mimic it since all its components influence aquatic life, and therefore fluvial ecosystems. It has been widely
described that human activities and climate change modify the natural hydrological regime. These changes in non-permanent
rivers generally tend towards greater intermittency, a condition that limits the applicability of hydrological alteration indices.
The general aim of the paper was to develop an aggregated impact index, the Hydrological Regime Index (HRI) suitable for
flow alteration assessment in non-permanent rivers. The HRI is composed of the flow magnitude attenuation, timing of
maximum flow and interannual flow variation impact factors. The HRI is based on simple conceptualisations and uses monthly
flow data, allowing its applicability in basins with limited information. The HRI was suitable to evaluate the impacts on the
river regime of both the Desaguadero-Salado-Chadiluevú-Curacó River which is severely dammed with intermit runoff and
the Colorado River with permanent runoff. In all the cases, the HRI was able to discriminate different impacts on the
hydrological regime for natural, low and high impoundment conditions. Thus, the HRI constitutes a very useful tool for
determining E-Flows and quantifying impacts due to water or land use changes.

## 19 1 Introduction

River networks expand and contract in response to the hydrological regime. Hydrological expressions can manifest in one or
all of the four dimensions, longitudinal, transversal, vertical, and temporal. These dimensions define the connectivity of the
fluvial system throughout the basin (Stanley et al., 1997; Amoros and Bornette, 2002; Gordon et al., 2004; Doering et al.,
2007). The hydrological regime of a river can in general terms, and despite other considerations, be defined about how the
flows are distributed throughout the year. A main consideration is whether the flows are permanent, intermittent, and/or
ephemeral (Sauquet et al., 2021).
Arid and semi-arid basins typically present intermittent runoff in some sectors of the drainage network. This intermittence can
be of different duration and extension (Datry et al., 2014; Boulton et al., 2017; Tramblay et al., 2021). In large basins, the
headwaters of the drainage network are generally located in a mountainous sector that favours the occurrence of precipitation
due to the orographic effect. Consequently, the hydrological forcing of the basin typically occurs in the headwaters and almost
none is manifested in the lower part. Moreover, higher temperatures result in important evapotranspiration losses which





accentuate the hydrological deficit of the lower part of the basin. Therefore, runoff is made up of allochthonous flows. Between
these events and depending on whether there is groundwater discharge that maintains a base flow, the riverbeds can dry up.
Snow-fed rivers present a well-defined hydrological regime in terms of flow timing and magnitude, with a pronounced peak
flow when snow is melting and low winter flow during the snow accumulation phase. However; all these hydrological
expressions are strongly modified by flow regulation, usually by the construction of dams to supply water for multiple uses
such as irrigation, recreation, domestic and hydroelectric generation (Magilligan et al., 2013). These effects are accentuated
by the use of low-efficiency irrigation systems, such as gravity-fed surface irrigation practices (McMahon and Finlayson, 2003;
Masseroni et al., 2017) and contribute to the loss of basin hydrological connectivity.
In addition, the human-caused impact on the hydrological regime of snow-fed rivers caused by the damming of large reservoirs
may be greater than the impact of climate change (Arheimer et al., 2017). This poses a challenge to the need to define
environmental flows (E-Flows). Regardless of the large number of approaches and methods for estimating E-Flows, there is a
consensus that E-Flows must mimic the hydrological regime, due to its structural and functional role in fluvial ecosystems
(Richter et al., 1996; Poff et al., 1997). In this sense, hydrological methods that include the description of the natural
hydrological regime are the most used (Arthington, 2012). However, knowing how the hydrological regime is affected is also
an essential factor in the definition of E-Flows using holistic approach methods. Moreover, tools for defining E-Flows must
be developed in transboundary fluvial systems that have fragmented water governance (Wineland et al., 2021).
The resulting major disturbances of flow regulation on the hydrological regime may include changes in the magnitude of flows
(i.e. flow attenuation), time delay of peak flows, loss of intra-annual variability, and reduction or loss of the hydrological
connectivity in the basin (Callow and Smettem, 2009; Steward et al., 2012; Magilligan et al., 2013; Torabi Haghighi et al.,
2014). Hydropower and flood management typically reduce flow variability and can affect the timing of peak flows, while
irrigation management usually reduces flow magnitude due to crop water use.
Several conceptualizations and metrics have been proposed to assess the effects of dams on the hydrological regime (e.g.
Richter et al., 1996, 1997 and 1998; Olden and Poff, 2003; Magilligan and Nislow, 2005; Poff et al., 2007; Gao et al., 2009;
Radinger et al., 2018; Döll and Schmied, 2012; Richter et al., 2012; Torabi Haghighi and Kløve, 2013; Torabi Haghighi et al.,
2014; Singh and Jain, 2020; Zhou et al., 2020; Sauquet et.al., 2021; Arthington, 2022; De Girolamo et al., 2022; McManamay
et al., 2022; Wang et al., 2022). However, in semi-arid regions the usual scarcity of data, such as the lack of detailed and
distributed information (e.g., discontinuous flow records and lack of daily data), and the intermittent flow conditions, limit the
use of flow alteration assessment indices (Leone et al., 2023; Gómez-Navarro et al, 2024). Indeed, indices based only on flow
statistics, such as the interquartile variation range, the coefficient of variation or the flow duration curve, used as proxies for
the seasonality of flows, among others; fail when no flow conditions are present. Therefore, new approaches to evaluate the
modification of hydrological regimens in non-perennial rivers are needed.
In this context, the Desaguadero Salado Chadileuvú Curacó (DSCC) River provides a representative case study because it is
an extensive semi-arid basin severely dammed which has undergone noticeable changes in its hydrological expression over
the past century mainly due to the fragmented water governance along its transboundary water systems (Dornes et al., 2016).





The fluvial system of the DSCC river develops over an extensive basin, with a highly heterogeneous relief, where winter
snowfall in the mountain area constitutes the main hydrological input function with a variability strongly influenced by the El
Niño-Southern Oscillation (ENSO) climate pattern (Compagnucci and Vargas, 1998; Compagnucci and Areneo, 2007;
Montecinos and Aceituno, 2003; Masiokas et al., 2006; Prieto et al., 2001; Araneo and Companucci, 2008; Barros et al., 2008;
Cortés et al., 2011; Penalba and Rivera, 2016; Rivera et al., 2017; Lauro et al., 2019).
This configuration determines a complex and non-linear hydrological basin response, which is modified by high impoundment
conditions. Thus, those years characterized as the warm or positive phase of ENSO (El Niño) led to heavy snowfall and above-
normal runoff that may exceed the storage capacity of the reservoirs, have less effect on the hydrological regime downstream
the reservoirs and a greater basing connectivity is observed. On the contrary, years characterised as the negative phase of
ENSO (La Niña) result in less snowfall and lower than normal streamflow which strongly modify the hydrological regime
downstream since almost no flow exceeds the storage capacity of the reservoirs, hence flows do not activate the lower part of
the DSCC River basin.
Since the flow regime is an integrated basin response, a comprehensive approach should be used to evaluate its temporal and
spatial distribution under both permanent and no-permanent flow conditions in areas with data scarcity. Therefore, to fill this
gap where many metrics did not properly evaluate the hydrological regime changes under non-flow conditions, the objective
of this study was to investigate the effect of flow regulation on the hydrological river regime by the development of a single,
simple and dimensionless index that can be applied in different regimes but especially under non-flow conditions with low
data requirement.
**2 Study Area**
The DSCC River basin constitutes the largest basin fully developed in the Argentine territory. The basin belongs to the
Colorado (CO) River that drains into the Atlantic Ocean (Figure 1). The DSCC River basin is located in the central-west part
of Argentina lying to the east of the Andes mountain range with a north–south orientation (27° 47' S, 38° 50′ S). It encompasses
partial or totally the provinces of Catamarca, La Rioja, San Juan, Mendoza, San Luis and La Pampa. The total area is
approximately 315,000 km$^2$ and includes the sub-basins of the Vinchina-Bermejo (VB), Jáchal (JL), San Juan (SJ), Mendoza
(MZ), Tunuyán (TY), Diamante (DT) and Atuel (AT) rivers. The DSCC River is located in the Andean piedmont and is defined
by mountain ranges such as the Cordillera de los Andes (CA) to the West and North, which includes the Cordillera Principal,
the Cordillera Frontal and the Precordillera, the Sierras Orientales and Sierras Pampeanas to the East, whereas the lower basin
is developed on flat terrain as part of the occidental area of the Pampean region (Ramos, 1999). Due to this orographic
configuration, the basin is isolated from the influences of wet air masses driven by the extratropical high-pressure systems of
the Atlantic and Pacific Oceans, a condition that results in an arid climate to the North and semiarid to the South (Prohaska,
1976). These conditions generate a north-south precipitation gradient that ranges from values around 100 to 350 mm per year
respectively, which does not contribute to the average hydrological expression of the DSCC River (Dornes et al., 2016).





The tributaries drain the eastern slope of the CA through well-defined valleys and canyons towards the piedmont. Tributary
streams reach their confluence with the DSCC River usually through depositional sediments forming alluvial fans where the
reduction of the terrain slope and the discharge of alluvial local aquifers, led to the occurrence of extensive wetlands. The
DSCC River initiates at the junction of the VB and SJ rivers, and the MZ River through the last one, as the outlet of the Lagunas
de Guanacahe (LG) wetland. It follows a North-South trajectory along approximately 1.450 km till its mouth in the CO River
at the Pichi Mahuida point in La Pampa province (38° 49′ S and 64° 59′ W). The DSCC River is distinguished by being an
axial collector that receives on its right bank all its tributaries forehead mentioned and connecting important wetlands
(Bereciartua et al., 2009, Chiesa et al., 2015), such as LG, Bañados del Tunuyán (BT), Bañados del Atuel (BA) and Lagunas
de Puelches (LP). Between these wetlands and until its mouth into the CO River, the DSCC River has different names. Thus,
it is called Desaguadero River (DSCC-I) between LG and BT, Salado River (DSCC-II) between BT and BA, Chadileuvú River
(DSCC-III) between BA and LP, and Curacó River (DSCC-IV) from LP to the CO River.

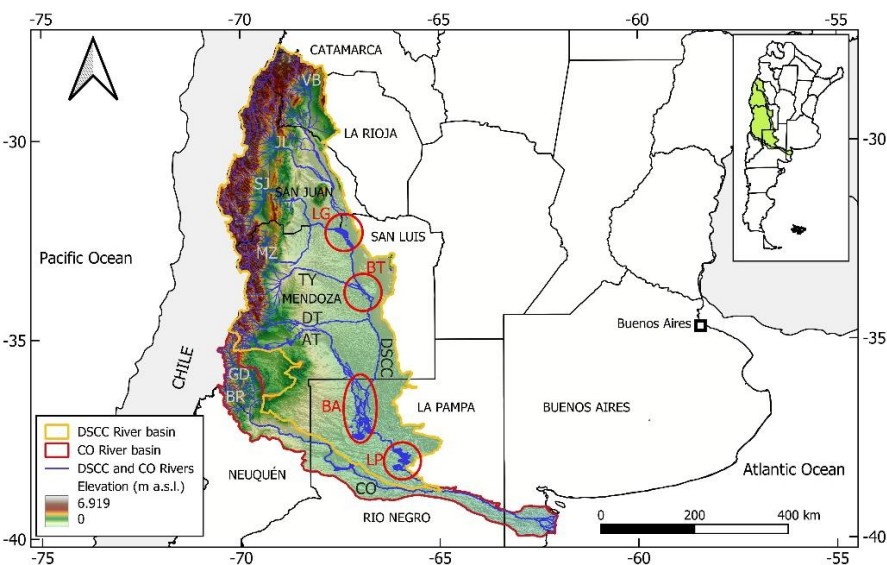

**Figure 1: Location of the Desaguadero-Salado-Chadileuvú-Curaco (DSCC) and Colorado (CO) River basins. VB: Vinchina-Bermejo River, JL: Jachal River, SJ: San Juan River, MZ: Mendoza River, TY: Tunuyán River, DT: Diamante River, AT: Atuel River, GD: Grande River, and BR: Barrancas River. Circles and ellipse indicate main wetlands: Lagunas de Guanacache (LG), Bañados del Tunuyán (BT), Bañados del Atuel (BA), and Lagunas de Puelches (LP).**

The wetlands of the DSCC River are epigenic as a result of the fluvial contributions with null groundwater discharge. They
are characterized by extensive flooded areas with numerous channels and lagoons, and acquire an ecological relevance due to
their location in a semi-arid region and for being hydrological regulation nodes of the basin. The LG, BT, and BA wetlands
are located at the distal part of extensive alluvial fans developed at the confluence of the corresponding tributary with the
DSCC River, therefore their hydrological expression depends more on the flow contribution of the tributary than on the DSCC
River. On the other hand, the LP wetland is characterized by the presence of extensive lagoons (e.g. La Brava, La Leona, La
Julia, La Dulce, Urrelauquen, and La Amarga) all of them linked by the DSCC River.





The headwater of the DSCC River basin is the CA, where winter precipitation due to the orographic lifting of Pacific air masses
by the mountains, constitutes the principal hydrological forcing of the basin. As a result, all the tributaries have a defined
snow-fed hydrological regimen, given that neither the glacier cover at the middle CA is significant nor the summer
precipitation. Northern sub-basins have considerably less runoff than the central and southern sub-basins as is the case of the
VB River with a mean discharge value around 1 $m^3 s^{-1}$, and JL River with an average annual flow of 10 $m^3 s^{-1}$. The SJ River
is the tributary with the greatest discharge with a mean annual flow of 65 $m^3 s^{-1}$ as a consequence of the development of the
basin over a large part of the CA covering a mountain front of more than 200 km. It is followed by the MZ River with 44 $m^3$
$s^{-1}$, whereas the TY, DT, and AT have 27, 31, and 34 $m^3 s^{-1}$ respectively (Table 1).
**Table 1: Gauging stations (GS) in the Desaguadero-Salado-Chadileuvú-Curacó (DSCC) and Colorado (CO) Rivers. [H]: historical**
**period, [A]: actual period, [N]: natural flow, [M]: modified flow. VB: Vinchina-Bermejo River, JL: Jáchal River, SJ: San Juan**
**River, MZ: Mendoza River, TY: Tunuyán River, AT: Atuel River, GD: Grande River, and BR: Barrancas River. VIN: Vinchina,**
**PAC: Pachimoco, PLT: Paso las Tunitas, EEN: El Encón, GUI: Guido, VDU: Valle de Uco, LJA: La Jaula, MCO: Monte Comán,**
**ESO: El Sosneado, CAA: Cañada Ancha, LAN: La Angostura, CAR: Carmensa, PTU: Puesto Ugalde, ADD: Arcos del Desaguadero,**
**SLT: Salto de la Tosca, CAN: Canalejas, STI: Santa Isabel, LRF: La Reforma, PUE: Puelches, PM2: Pichi Mahuida 2, LGR: La**
**Gotera, BAR; Barrancas, BRQ: Buta Ranquil, and PMA: Pichi Mahuida (PMA).**

| River | Sub-basin | GS | ID | Lat S | Long. W | Elevation (m a.s.l.) | Mean annual Discharge ($m^3 s^{-1}$) | Record period |
|---|---|---|---|---|---|---|---|---|
| DSCC | VB | VIN | 1001 | 28.75 | 68.25 | 1480 | 1.3 [H, N] | 1967-1981 |
|  |  |  |  |  |  |  | 0.4 [A, M] | 2016-2023 |
|  | JL | PAC | 1204 | 30.21 | 68.83 | 1160 | 14.6 [H, N] | 1921-1928 |
|  |  |  |  |  |  |  | 9.6 [H, N] | 1936-1990 |
|  | SJ | km 43.7 | 1208 | 31.52 | 68.94 | 934 | 65.2 [H, N] | 1909-2014 |
|  |  | km 101 | 1211 | 31.25 | 69.18 | 1245 | 55.6 [H, N] | 1971-2005 |
|  |  |  |  |  |  |  | 30.7 [A, N] | 2010-2023 |
|  |  | PLT | 1408 | 32.12 | 68.16 | 531 | 16.8 [H, M] | 1937-1951 |
|  |  | EEN | 1219 | 32.23 | 67.81 | 518 | 11.8 [H, M] | 1993-2023 |
|  |  |  |  |  |  |  | 0.9 [A, M] | 2010-2023 |
|  | MZ | GUI | 1413 | 39.92 | 69.24 | 1408 | 43.6 [H, N] | 1956-2023 |
|  |  |  |  |  |  |  | 32.8 [A, N] | 2010-2023 |
|  |  | CAC | 1412 | 33.02 | 69.12 | 1250 | 50.2 [H, N] | 1909-1990 |
|  | TY | VDU | 1419 | 33.78 | 69.27 | 1199 | 27.0 [H, N] | 1954-2023 |
|  |  |  |  |  |  |  | 17.5 [A, N] | 2010-2023 |
|  | DT | LJA | 1423 | 34.67 | 69.32 | 1457 | 31.2 [H, N] | 1971-2023 |
|  |  |  |  |  |  |  | 19.1 [A, N] | 2010-2023 |
|  |  | MCO | 1451 | 34.57 | 67.87 | 521 | 7.5 [H, M] | 1990-2023 |
|  |  |  |  |  |  |  | 3.0 [A, M] | 2010-2023 |
|  | AT | ESO | 1428 | 35.08 | 69.60 | 1603 | 36.0 [H, N] | 1972-2023 |
|  |  | CAA | 1415 | 35.19 | 69.78 | 1680 | 9.4 [H, N] | 1940-2023 |
|  |  | LAN | 1403 | 35.10 | 68.87 | 1302 | 34.4 [H, N] | 1906-2023 |
|  |  |  |  |  |  |  | 24.0 [A, N] | 2010-2023 |
|  |  | CAR | 1453 | 35.19 | 37.73 | 438 | 7.1 [H, M] | 1985-2023 |
|  |  |  |  |  |  |  | 3.9 [A, M] | 2010-2023 |
|  |  | PTU | 4404 | 36.00 | 67.19 | 343 | 6.6 [H, M] | 1980-2023 |
|  |  |  |  |  |  |  | 2.0 [A, M] | 2010-2023 |
|  | DSCC | ADD | 1424 | 33.40 | 67.15 | 450 | 15.9 [H, M] | 1941-1951 |
|  |  |  |  |  |  |  | 0, 1[A, M] | 2010-2023 |
|  |  | SLT | 1605 | 34.09 | 66.71 | 404 | 5.1 [H, M] | 1944-1950 |
|  |  |  |  |  |  |  | 0.2 [A, M] | 2017-2023 |
|  |  | CAN | 1452 | 33.17 | 66.50 | 356 | 13.0 [H, M] | 1987-2023 |
|  |  |  |  |  |  |  | 1.1 [A, M] | 2010-2023 |
|  |  | STI | 4403 | 36.28 | 66.85 | 310 | 37.5 [H, M] | 1980-2023 |
|  |  |  |  |  |  |  | 1.2 [A, M] | 2010-2023 |
|  |  | LRF | --- | 37.55 | 66.23 | 243 | 30.2 [H, M] | 1980-2023 |
|  |  |  |  |  |  |  | 0.4 [A, M] | 2010-2023 |





| | | PUE | --- | 38.15 | 65.91 | 222 | 22.2 [H, M] | 1982-2023 |
|----|-----|-----|------|-------|-------|------|-------------|-----------|
| | | | | | | | 0.0 [A, M] | 2010-2023 |
| | | PM2 | --- | 38.82 | 64.99 | 125 | 12.0 [H, M] | 1982-2023 |
| | | | | | | | 0.0 [A, M] | 2010-2023 |
| CO | GR | LGT | 1427 | 35.87 | 69.89 | 1454 | 100.2 [H, N] | 1973-2023 |
| | BR | BAR | 2001 | 36.80 | 69.89 | 950 | 34.0 [H, N] | 1960-2023 |
| | CO | BRQ | 2002 | 37.07 | 69.74 | 850 | 140.9 [H, N] | 1940-2023 |
| | | | | | | | 79.1 [H, N] | 2010-2023 |
| | | PMA | 1801 | 38.82 | 64.98 | 122 | 133.6 [H, N] | 1918-1990 |
| | | | | | | | 59.3 [A, M] | 2010-2023 |


All the tributaries show both a great interannual flow variability that is consistent with varying snowmelt processes occurring
in a complex mountain environment and a defined synchronicity with above and below-average flows strongly related to
positive and negative ENSO episodes (Compagnucci and Vargas, 1998, Aceituno and Vidal, 1990; Waylen and Caviedes,
1990; Masiokas et al., 2006; Araneo and Villalba, 2014). The maximum flow magnitudes observed in 1980s, 1992, 1995,
2005, and 2006 and to a lesser degree in 2008 were associated with El Niño episodes. On the opposite, the last decade showed
very low flow values, according to the dominance of negative ENSO phases (La Niña), with the exclusion of 2015 classified
as an El Niño episode that resulted in average flow values. As a consequence, lesser natural flows are seen in all the tributaries
for the actual conditions
The DSCC River basin has twelve large reservoirs; all located on its tributaries (Figure 2 and Table 2). Currently, El Tambolar
(ETA) on the SJ River is under construction and there is more planned such as El Baqueano (EBA) on the DT River. None of
them were built for flood control; instead, they were built for irrigation purposes and hydropower generation. The prevalent
use of inefficient gravity-fed surface irrigation systems determines that irrigation demands are unusually high with respect to
natural supply. As a result of these impoundments and reservoir operation, none of the tributaries contributes in natural regimen
to the DSCC River. Further, in the DSCC River, two small dams (Azud Norte, AZN, and Azud Sur, AZS) were built to generate
impoundment conditions and prevent erosion in the LG wetland. The CO River, has the Dique Punto Unido (DPT) diversion
dam used for irrigation and water consumption, and the Casa de Piedra (CDP) reservoir that regulates the different water
allocations in the lower basin.





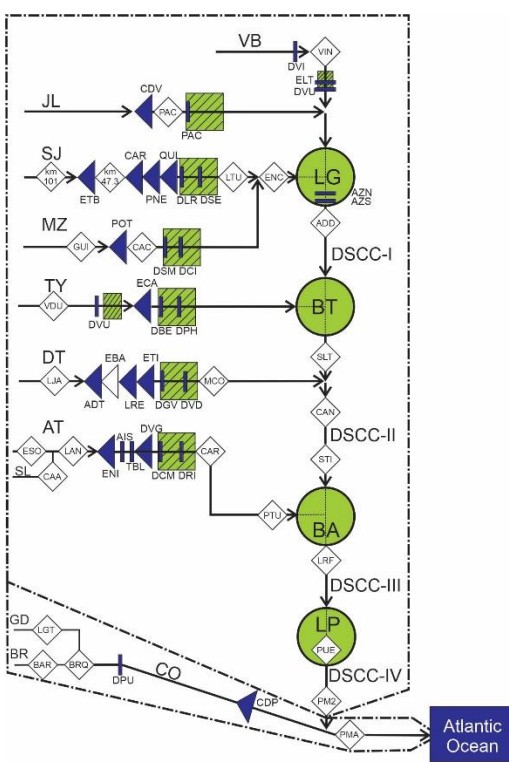

**Figure 2: Schematic diagram of the Desaguadero-Salado-Chadileuvú-Curacó (DSCC) River and Colorado (CO) River basins showing the location of its reaches, tributaries, wetlands, gauging stations, reservoirs, diversions dams and irrigated areas. DSCC-I: Desaguadero River, DSCC-I: Salado River, DSCC-III: Chadileuvú River, and DSCC-IV: Curacó River. Circle: wetlands. Dark triangles: main reservoirs. White triangle: projected reservoir. Rectangles: diversion or flood control dams. Shaded squares: irrigation areas. Diamonds: gauging stations. More descriptions are depicted in Figure 1, and Tables 1 and 2.**

High flow records are strongly associated with El Niño episodes, such as in the 1980s decade when the DSCC River drainage network was fully active with discharges to the CO River. The runoff in the DSCC River is allochthonous due to the reduced rainfall that dominates the lower basin. The historical information is not synchronous, given that it is generally only available in periods with runoff, reveals highly modified and severely attenuated annual hydrographs along the DSCC River. The actual situation shows an even more drastic hydrological condition with almost no flow in all its extension. Thus, as a consequence of the described flow regulation in the tributaries, the DSCC River is actually dry. Furthermore, no groundwater discharge is observed from outside the alluvial plain. Groundwater flow follows the regional gradient of the river and it is majorly constrained to the alluvial plain of the DSCC River where the phreatic aquifer is fed by fluvial recharge (Páez Campos and Dornes, 2021).

**Table 2: Subbasins, reservoirs and diversion dams in the Desaguadero-Salado-Chadileuvú-Curacó (DSCC) River basin and in the Colorado (CO) River basin. Elevation obtained from the Digital Elevation Model (DEM). *Diversion and flood control dam. DVI: Dique Vinchina, ELT: Embalse Lateral, DVU: Dique Villa Unión, CDV: Cuesta del Viento, PAC: Pachimoco, ETA: El Tambolar, CAL: Caracoles, PTN: Punta Negra, QUL: Quebrada de Ullúm, DLR: Dique La Rosa, DSE: Dique San Emiliano, POT: Potrerillos, DSM: Dique San Martín, DCI: Dique Cipolletti, DVU: Dique Valle de Uco, ECA: El Carrizal, DBE: Dique Benegas, DPH: Dique Phillps, ADT: Agua del Toro, LRE: Los Reyunos, ETI: El Tigre, DGV: Dique Galileo Vitali, DVI: Dique Vidalino, ENI: El Nihuil,**





**AIS: Aisol, TBL: Tierras Blancas, VGR: Valle Grande, DCM: Dique Canal Marginal, DRI: Dique Rincón del Indio, AZN: Azud**
**Norte, AZS: Azud Sur, Dique Punto Unido (DPU) and CDP: Casa de Piedra.**

| River | Subbasin | Area (km²) | Max. Elevation (m a.s.l.) | Min. Elevation (m a.s.l.) | Reservoirs and diversion dams | Vol Reservoirs (hm³) |
|---|---|---|---|---|---|---|
| DSCC | VB | 35,850.2 | 5,195 | 532 | DVI* | < 1 |
| | | | | | ELT* | < 1 |
| | | | | | DVU* | < 1 |
| | JL | 34,716.6 | 5,296 | 695 | CDV | 206 |
| | | | | | PAC* | < 1 |
| | SJ | 38,813.3 | 4,850 | 555 | ETA | 605 |
| | | | | | CAL | 565 |
| | | | | | PTN | 450 |
| | | | | | QUL | 440 |
| | | | | | DLR* | < 1 |
| | | | | | DSE* | < 1 |
| | MZ | 17,861.7 | 6,556 | 539 | POT | 180 |
| | | | | | DSM* | < 1 |
| | | | | | DCI* | < 1 |
| | TY | 21,384.2 | 4,766 | 476 | DVU* | < 1 |
| | | | | | ECA | 327 |
| | | | | | DBE* | < 1 |
| | | | | | DPH* | < 1 |
| | DT | 8,638.2 | 4,082 | 413 | ADT | 380 |
| | | | | | LRE | 255 |
| | | | | | ETI | 70 |
| | | | | | DGV* | < 1 |
| | | | | | DVI* | < 1 |
| | AT | 54,832.5 | 3,118 | 298 | ENI | 384 |
| | | | | | AIS* | < 1 |
| | | | | | TBL* | < 1 |
| | | | | | VGR | 164 |
| | | | | | DCM* | < 1 |
| | | | | | DRI* | < 1 |
| | DSCC | 102,842.4 | 1,612 | 214 | AZN* | 10 |
| | | | | | AZS* | 138 |
| **Total** | | **314,939.1** | **6,556** | **214** | | |
| CO | | 47,458.9 | 3,230 | 0 | DPU* | < 1 |
| | | | | | CDP | 400 |


The resulting lack of hydrological connectivity of the DSCC River with the upper basin where snowmelt runoff is generated
determines a strong hydrological deficit in the lower basin that has significant ecological effects. Indeed, the dominance of
evaporation processes results in a severe salinization of its environment and the lack of contribution to the CO River. Figure 3
illustrates the annual hydrographs for both the available historical information and the actual period (2010-2023) of the
tributaries and the DSCC River.



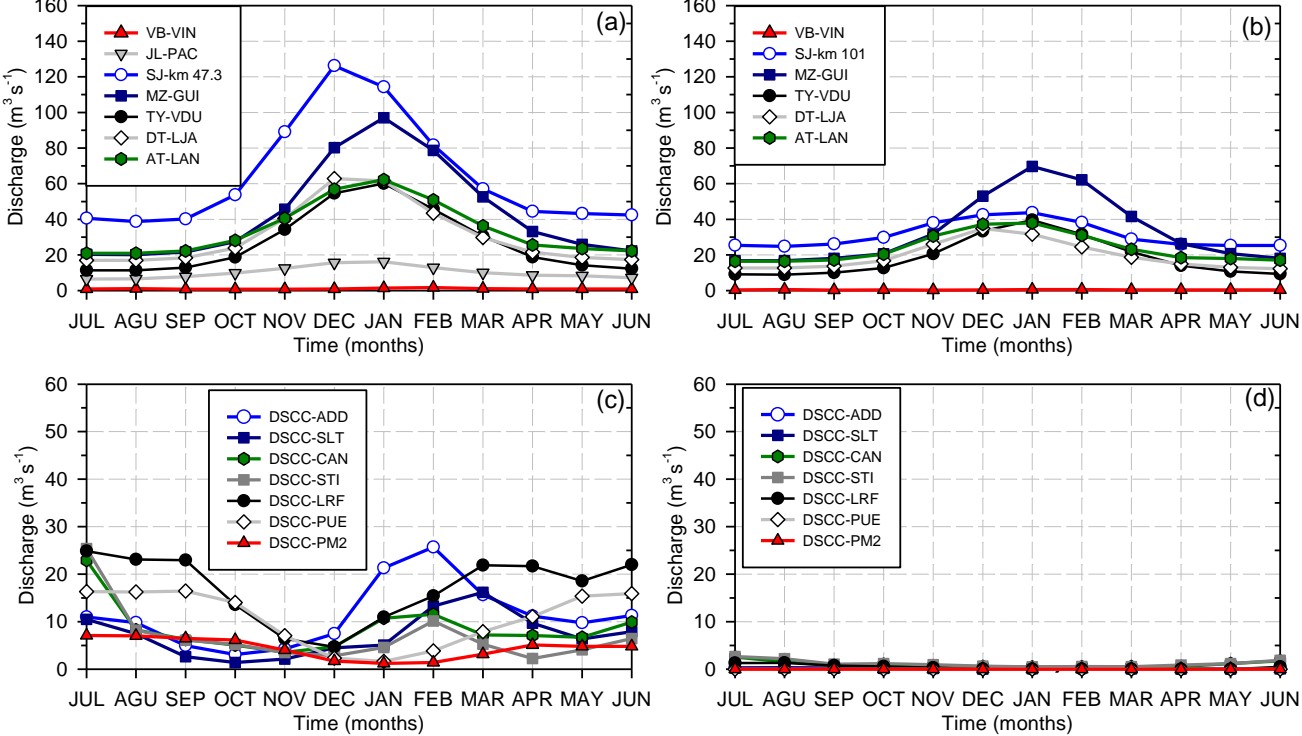

**184**

**Figure 3: Annual hydrographs of the Desaguadero-Salado-Chadileuvú-Curacó (DSCC) River. a) Historical flows in natural regime of the tributaries of the DSCC, b) Actual flows (2010-2023) in natural regime of the tributaries of the DSCC, c) Historical flows in modified regime of the DSCC, and d) Actual flows (2010-2023) in modified regime of the DSCC River. Rivers, gauging stations, and historical and actual periods detailed in Table 1.**

**3. Materials and methods**

**3.1 Development of the hydrological regime index**

To evaluate the effects of flow regulation on the hydrological river regime in different hydrological conditions but mainly in intermittent rivers, a single impact index, the Hydrological Regime Index (HRI) was developed. The HRI incorporates the main components of the hydrological regime (i.e. flow attenuation, time shifting of maximum flow and inter-annual variability). It is based on the comparison of the annual distribution of monthly flow records in natural and modified regimes (i.e. upstream vs downstream of a reservoir) which is by definition the hydrological regime. Since the HRI is not a site-specific



measurement, this approach makes it suitable for non-flow conditions and to evaluate the limitation or loss of hydrological
connectivity due to flow impoundment along the river network.
To facilitate the comparison of the flow records and similarly to the concept of the unit river approach used by Torabi Haghighi
and Kløve (2013), the flows are scaled to have an equal flow rate (U) of 100 million cubic meters (MCMs) per year. Therefore,
the scaled monthly flows ($Q_{sm}$) are calculated as the contribution to the annual flow following Eq. (1):
$$Q_{sm} = \frac{Q_m}{Q_a} \times U \qquad (1)$$
where Qm is the monthly flow and Qa is the annual average flow rate of the river. This scaling allows rivers with different
discharge rates to be compared in terms of the annual hydrological regime. A uniform regulated o dry river has a Qsm of
exactly 8.333 MCM of total flow.
Similar to the approach applied by Torabi Haghighi et al (2014), but using simpler functions adapted to intermittent flows to
describe the time lag and interannual variability, the HRI is detailed as follows in Eq. (2):
$$HRI = MIF \times (TIF + VIF) \qquad (2)$$
where *HRI* varies between 1 (natural or unmodified flow) and 0 (completely modified flow). *MIF*: Magnitude Impact Factor,
*TIF*: Timing Impact Factor, and *VIF*: Variation Impact Factor.
MIF is of equal importance to the sum of TIF and VIF because flow magnitude is the main controlling factor of the hydrological
regime. For example, for a no-flow condition, MIF is 0 and HRI must be 0 (i.e. completely modified flow). The maximum
impact of TIF or VIF is 0 and their sum is 1 when no changes in timing and intra-annual variability are observed.
Flows downstream of multipurpose reservoirs typically result in values of lower magnitude due to different water consumption.
The extreme cases are when there are no downstream flows (MIF=0) or when the upstream and downstream flows are equal
(MIF=1). Since MIF is calculated based on average values over long o representative periods, is very rare to have larger flow
values downstream of a reservoir. However, if this is the case, MIF is set equal to 1. Therefore, MIF was calculated as the ratio
between modified to natural flows as in the following Eq. (3):
$$MIF = Q_{aM}/Q_{aN} \qquad (3)$$
where $Q_{aM}$ is the mean annual modified flow (e.g. downstream of the reservoir) and $Q_{aM}$ is the mean annual flow in natural
regime (e.g. upstream of the reservoir).
The TIF was calculated based on the time delay in monthly maximum discharge (TD) along the hydrological year between the
natural (e.g. upstream of the reservoir) and modified flows. The maximum TD value is 6 months corresponding with a seasonal
inverted maximum flow, therefore the following conditionals are applied in Eq. (4) and (5):
$$\text{if } TD = |TQ_{mN.max} - TQ_{mM.max}| \leq 6; \ TD = |TQ_{mN.max} - TQ_{mM.max}| \qquad (4)$$
$$\text{if } TD = |TQ_{mN.max} - TQ_{mM.max}| > 6; \ TD = 12 - |TQ_{mN.max} - TQ_{mM.max}| \qquad (5)$$





where $TQ_{mN.max}$ and $TQ_{mM.max}$ are the time (i.e. number of months) of occurrence of the monthly natural and modified maximum
flow respectively.
To scale the TIF to a maximum value of 0.5 (i.e. natural flow) and a minimum value of 0 (i.e. maximum time shift), the TD is
calculated as following Eq. (6):
$TIF = 0.5 - 0.0833 \times TD$                                                     (6)
Regardless of the type and operation of the reservoir, the resulting downstream flow is more uniform, which represents a loss
of interannual variability. Complete regulation implies a constant flow rate, which can be equal to the average annual flow rate
or have a lower value up to a flow rate equal to zero. Therefore, the VIF is calculated based on the annual sum of the deviations
from a straight or constant flow line for both the natural and modified flow. These values are the Monthly Regime Index (MRI)
and are totalized in the Annual Regime Index (ARI). Both, the MRI and ARI are computed using the scaled hydrographs (Eq.
1), therefore if the Qm is constant (i.e. uniform regime); the Qsm= 8.333, and MRI=ARI=0. The following conditions are
applied in Eq. (7, 8 and 9):
If $Q_{sm} = 8.333; MRI = 0$                                                     (7)
If $Q_{sm} > 8.333; MRI = |Q_{sm} - 8.333|$                                         (8)
$ARI = \sum_{i=1}^{12} MRI_i$                                                       (9)
The Annual Regime Index for natural flows ($ARI_N$) typically varies between 30 to 55 for snow-fed regimes. Modified flows
can have values of the Annual Regime Index ($ARI_M$) between 0 (i.e. equal value all the months) and a maximum value of 91.67
when a dry river has runoff occurring only in one month (i.e. ephemeral river). To scale the VIF between 0.5 (i.e. natural
flows) and a minimum value of 0 (i.e. maximum flow regime modification) the Relation Regime Index (RRI) between the
natural and modified flows is defined in Eq. (10):
$RRI = ARI_M / ARI_N$                                                           (10)
The following conditions must be considered. If the observed annual flow variability downstream is lower than the one
upstream (i.e. RRI<1), the RRI value is scaled so that VIF varies between 0 and 0.5. On the contrary, if the flow variability
downstream is larger than the one upstream (i.e. RRI>1) it means that a drastic modification occurred to the streamflow given
by dam management or by the contribution of no natural flow such as drainage discharges from irrigation areas. In this case,
VIF equals 0. To avoid a drastic change between values of RRI=1 (VIF 0.5) and RRI>1 (VIF=0) a transition function was
introduced to consider an increase in the non-natural variability of less than 20% as indicated in the following Eq. (11, 12 and

253   13):

If $0 < RRI \leq 1; VIF = 0.5 \times RRI$                                           (11)
If $1 < RRI \leq 1.2; VIF = -2.5 \times RRI + 3$                                     (12)





If $RRI > 1.2$; $VIF = 0$ (13)
Finally, seven different impact classes were defined for different values of HRI using percentiles as indicated in Table 3. The
two classes at the lower and upper extremes have an extension of 10% in relation to the 20% that the middle classes present.
This was implemented to highlight severe and drastic impacts or low impact conditions respectively.
**Table 3: Hydrological Regime Index (HRI) impact classes**

| Range | Impact class |
|---|---|
| $0.0 \leq HRI < 0.1$ | Drastic |
| $0.1 \leq HRI < 0.2$ | Severe |
| $0.2 \leq HRI < 0.4$ | High |
| $0.4 \leq HRI < 0.6$ | Moderate |
| $0.6 \leq HRI < 0.8$ | Incipient |
| $0.8 \leq HRI < 0.9$ | Low |
| $0.9 \leq HRI < 1.0$ | Extremely Low |

**3.2 Data set**
The HRI was applied in the DSCC river basin, which is currently characterized by its hydrological discontinuity and
intermittent flows. Therefore, natural flows were evaluated in the tributaries upstream the main reservoirs, whereas modified
flows downstream the main reservoirs were analyzed by comparing them with flow records registered upstream. Similarly, in
the lower basin of the DSCC River and given the great distance from the reservoirs, the modified flows were analyzed by
comparing periods of flows in natural regime with others in modified regime under low and high impoundment conditions.
Moreover, to validate the applicability of the index, the HRI was also applied to the CO River with a defined hydrological
connectivity throughout the basin and permanent runoff in natural regimen, and with both low and high impoundment
conditions.
In the tributaries of the DSCC River, the HRI was calculated on those rivers with flow in natural regime by comparing at least
two gauging stations upstream of the main reservoirs. The gauging stations were selected for their proximity, to ensure that
there are no significant contributions from streams or interactions with groundwater. In the case that the distances are greater,
the criterion was based on the allochthonous nature of the flows, that is, there are no obvious contributions in the analyzed
section that result in greater flows at the downstream gauging station. Based on the above and the availability of information,
the MZ River at GUI and CAC (1956-90) and AT River at ESO plus the contribution of the Salado (SL) River respect to the
records downstream in LAN (1972-03), were evaluated. In the CO River basin, the HRI for natural flows was implemented in
the headwaters (LGT and BAR) with respect to the monthly flows registered in BRQ, and in the main channel between BRQ
and PMA gauging stations, for the 1976-2011 and 1940-1971 periods respectively (Table 4)
**Table 4: Detail of the gauging stations (GS) located upstream [us] and downstream [ds] of each other and periods with common data**
**available used to calculate the Hydrological Regime Index (HRI) for natural flows in the tributaries of Desaguadero-Salado-**
**Chadileuvú-Curacó (DSCC) River and in the Colorado (CO) River. Mendoza (MZ) River at Guido (GUI) and Cacheuta (CAC),**
**Atuel (AT) River at El Sosneado, Salado (SL) River at Cañada Ancha (CAA), and AT River at La Angostura (LAN), Grande (GD)**
**River at La Gotera (LGT), Barrancas (BR) River at Barrancas (BAR), CO River at Buta Ranquil (BRQ) and Pichi Mahuida (PMA).**





| River | Tributaries | GS [us] | GS [ds] | period |
|-------|-------------|---------|---------|--------|
| DSCC | MZ | GUI | CAC | 1956-1990 |
|  | AT + SL | ESO + CAA | LAN | 1972-2023 |
| CO | GD + BR | LGT + BAR | BRQ | 1976-2011 |
|  |  | BRQ | PMA | 1940-1971 |


Further, to analyse the HRI performance in evaluating the impact of reservoirs on flow conditions, the HRI was applied in the
DSCC River basin in two sectors, the tributaries and the lower reaches of the DSCC River, based on flow data availability.
The effect of the reservoirs and their operation on the hydrological regime was contemplated for different impoundment
conditions by comparing the flow in gauging stations located downstream of the reservoirs with the upstream records in the
natural regime (Table 5). In this case, only in the SJ River (km 47.3 vs PLT) was possible to evaluate the effect of a low
impoundment condition from 1937 to 1950 and in the CO river (BRQ vs PMA) for the 1940-1971 period. For the current
impoundment conditions, the modification of the hydrological regime was analysed in the majority of the tributaries of the
DSCC River (SJ, MZ, DT and AT) in two periods, the historical available records till 2010 and the 2010-2023 time series that
represent both the actual impoundment and climate conditions. In the SJ River, the sum of natural flows at SJ-km 47.3 or SJ-
km 101 and in the MZ River at MZ-GUI were compared with those observed in SJ-EEN (modified flow) for the two indicated
periods extending from 1993 to 2023. In the DT River, the natural flows at DT-LJA were compared with modified flows
recorded in DT-MOC for the historical and actual periods, while in the AT River the natural flows at AT-LAN were contrasted
with the modified flows registered at AT-CAR y AT-PTU for the 1985-2023 and 1980-2010 time series respectively splitting
the analyses in the two previously indicated periods.
Similar approach was applied in the CO River, where for low impoundment conditions natural monthly flows recorded in BRQ
were compared with the modified observed in PMA for the 1972-1990 period. For high impoundment conditions, flows
recorded BRQ and PMA were contrasted for the available historical (1994-2010) and actual (2010-2023) periods. Missing
records in PMA between 2015-2018 and 2023 were completed with CDP flow discharges while the flow contributions of the
DSCC River in the 1980s were subtracted.
**Table 5: Detail of the gauging stations (GS) located upstream [us] and downstream [ds] of reservoirs and periods with common
available data used to calculate the Hydrological Regime Index (HRI) for modified flows in the Desaguadero-Salado-Chadileuvú-
Curacó (DSCC) River and the Colorado (CO) River. San Juan (SJ) River at km 47.3, km 101, Paso las Tunitas (PLT) and El Encón
(EEN), Mendoza (MZ) River at Guido (GUI), Diamante (DT) River at La Jaula (LJA) and Monte Comán (MCO), Atuel (AT) River
at La Angostura (LAN), Carmensa (CAR) and Puesto Ugalde (PTU), CO River at Buta Ranquil (BRQ), Pichi Mahuida (PMA) and
Casa de Piedra (CDP). [N] and [M], natural and modified flows, [*] and [+] low and high impoundment conditions, [H] and [A]
historical and actual conditions respectively.**

| River | Tributaries | GS [us, N] | GS [ds, M] | period |
|-------|-------------|------------|------------|--------|
| DSCC | SJ | Km 47.3 | LTU | 1937-1951 [H*] |
|  | SJ+MZ | Km 47.3 + GUI | EEN | 1993-2010 [H+] |
|  | SJ+MZ | Km 101 + GUI | EEN | 2010-2023 [A+] |
| DSCC | DT | LJA | MCO | 1990-2010 [H+] |
|  | DT | LJA | MCO | 2010-2023 [A+] |
| DSCC | AT | LAN | CAR | 1990-2010 [H+] |
|  | AT | LAN | CAR | 2010-2023 [A+] |
|  | AT | LAN | PTU | 1990-2010 [H+] |
|  | AT | LAN | PTU | 2010-2023 [A+] |





| CO | BRQ | PMA | 1972-1990 [H*] |
|----|-----|-----|----------------|
|    | BRQ | PMA | 1994-2010 [H+] |
|    | BRQ | PMA/CDP | 2010-2023 [A+] |

In the DSCC River, the lack of records with natural flows and the intermittence of the actual flow records determined that the application of the HRI was based on a temporal comparison. If 1988, which activated the entire fluvial system, is considered an approximate condition of the natural regimen, it is possible to compare it with current flow conditions (2010-2023). Logically, 1988 represented a year of extraordinary flows that result in greater attenuations when compared to the current flows. For this reason, the 1982-1992 time series was considered as reference period, since its records include both flooding and low-water years. As a result of data availability, the STI, LRF, PUE, and PM2 gauging stations located in the lower DSCC River basin were used.

The information is available at the national hydrological information system (SNIH) of the Secretaría de Infrestructura y Política Hídrica de Argentina, https://snih.hidricosargentina.gob.ar, in the hydrological database of La Pampa province (BDH) of the Secretaría de Recursos Hídricos de La Pampa, https://bdh.lapampa.gob.ar , and in the Colorado River Interjurisdictional Committee (COIRCO), https://www.coirco.gov.ar.

## 4. Results

### 4.1 Hydrological regime index in natural flow

The performance of the HRI was first evaluated for rivers with flow in natural regimes in both the tributaries of the DSCC River and in the CO River (Figure 4). In this case, the average monthly flows in natural regime recorded at a given gauging station were compared with those recorded upstream.

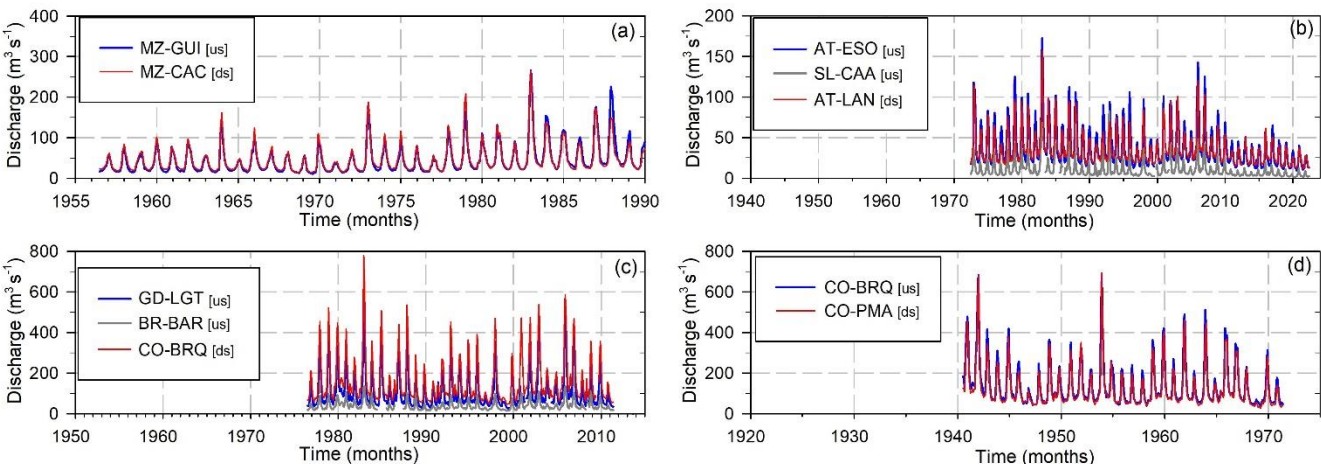

Figure 4: Chronological monthly flows in natural regime of the tributaries of the Desaguadero-Salado-Chadleuvú-Curacó (DSCC) River and in the Colorado (CO) River used to calculate the Hydrological Regime Index (HRI). [us]: upstream, [ds]: downstream. a)





**Mendoza (MZ) River at GUI and CAC, b) Atuel (AT) River at ESN, Salado (SL) River at CCA, and AT-LAN, c) Grande (GD) River at LGT, Barrancas (BR) River at BAR, and CO River at BRQ, and d) CO River at BRQ and PMA.**

For all the rivers analysed in natural regime, low HRI values were observed (Table 6 and Figure 5). In the MZ River, the comparison between flows recorded in GUI and CAC gauging stations for the period 1956-1990, previous to the construction of the POR reservoir, showed that there was no flow attenuation between GUI and CAC gauging stations. CAC had a slightly higher average annual flow value, possibly as a result of the contribution of streams between both stations, since they were located approximately 17.5 km from each other. Therefore, MIF was set equal to 1. There was no time delay (TIF=0.5) and a slightly lesser interannual flow variation was seen in CAC (VIF=0.476). The resulting HRI of 0.98 indicates an extremely low modification of the hydrological regime.

In the AT River, the analysis was carried out from the monthly flows recorded in ESO plus the contributions from its tributary the Salado (SL) River in CAA, and compared with the flow records in LAN located approximately 90 km downstream of both gauging stations. Both rivers join in the place called Las Juntas located at the foot of an extensive alluvial fan where significant flow losses occur and therefore lower flows are recorded in LAN. This resulted in an important attenuation of the flow magnitude (MIF= 0.785), however smaller impacts were seen in the timing and flow variability (TIF= 0.417 and VIF= 0.386). The HRI equals 0.63 and indicates an incipient modification of the hydrological regime.

In the headwater of the CO River basin, the monthly flows for the 1976-2011 period of the GD River in LGT plus those of the BR River in BAR were contrasted, with the flows recorded at the BRQ gauging station, located 160 and 37 km downstream respectively (see Figure 2). Due to contributions from small streams in the river section between the gauging stations analysed, the average annual flow is 5 % larger downstream in BRQ. Therefore, no flow attenuation was observed and the MIF equalled 1. In addition, no temporal differences were observed in the maximum flows (TIF=0.5) and a slightly lower interannual variability (VIF=0.475) was seen. The HRI equals 0.98 and shows that hydrological regime in natural conditions presented an extremely low modification between the analysed gauging stations. In the CO River, the monthly flows recorded in BRQ were compared with those of the PMA gauging station located 150 km downstream for the 1940-1971 period. Flows showed a low magnitude attenuation downstream that resulted in a MIF=0.883. The timing of maximum flows did not change (TIF=0.5) and the loss of interannual variability was very low (VIF=0.493). These impact factors resulted in a HRI=0.88 that indicates a low impact on the hydrological regime for the CO River in natural regime.

**Table 6: Hydrological Regime Index (HRI) for natural flows in the tributaries of the Desaguadero-Salado-Chadileuvú-Curacó (DSCC) River and in the Colorado (CO) River. Qma: mean annual flow. [us]: upstream, [ds]: downstream. MIF: Magnitude Impact Factor, TIF: Timing Impact Factor, and VIF: Variation Impact Factor. Mendoza (MZ) River at Guido (GUI) [us] and Cacheuta (CAC) [ds], Atuel (AT) River at El Sosneado (ESO) [us], Salado (SL) River at Cañada Ancha (CAA) [us], and AT River at La Angostura (LAN) [ds], Grande (GD) River at La Gotera (LGT) [us], Barrancas (BR) River at Barrancas (BAR) [us], CO River at Buta Ranquil (BRQ) [ds and us] and Pichi Mahuida (PMA) [ds].**

| River | Series | Qma (m³ s⁻¹) [us] | Qma (m³ s⁻¹) [ds] | MIF | TIF | VIF | HRI | Impact class |
|---|---|---|---|---|---|---|---|---|
| MZ | 1956-90 | 44.4 (GUI) | 46.2 (CAC) | 1 | 0.5 | 0.475 | 0.98 | Extremely Low |
| AT+SL | 1972-23 | 36 (ESO) + 9.5 (CAA) | 35,7 (LAN) | 0.785 | 0.417 | 0.386 | 0.63 | Incipient |
| GD+BR | 1976-11 | 111.1 (LGT)+39.2 (BAR) | 158.1 (BRQ) | 1 | 0.5 | 0.475 | 0.98 | Extremely Low |
| CO | 1940-71 | 136.3 (BRQ) | 120.4 (PMA) | 0.883 | 0.5 | 0.493 | 0.88 | Low |



364

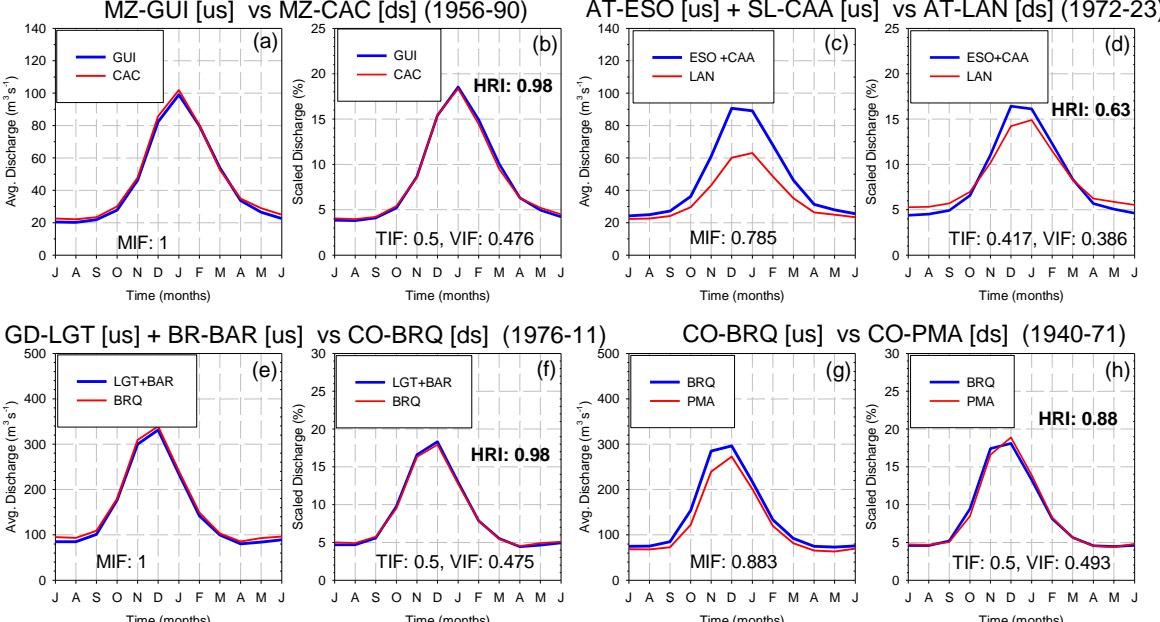

Figure 5: Hydrological Regime Index (HRI) for flows in natural regime in the tributaries of the Desaguadero-Salado-Chadileuvú-Curacó (DSCC) River and in the Colorado (CO) River. Annual and scaled hydrographs between gauging stations located upstream [us] and downstream [ds]. MIF: Magnitude Impact Factor, TIF: Timing Impact Factor, and VIF: Variation Impact Factor. a and b) Mendoza (MZ) River at GUI and CAC, c and d) Atuel (AT) River at ESO, Salado (SL) River at CAA, and AT River and LAN, e and f) Grande (GD) River at LGT, Barrancas (BR) River at BAR, and CO River at BRQ, and g and h) CO River at BRQ and PMA.

**4.2 Hydrological regime index with low impoundment conditions**

On the DSCC River basin, most of the main reservoirs were built on its tributaries in the second half of the 20th century. Previously, there were only small water diversion dams with little or no impoundment conditions (see Table 2). The present analysis is thus restricted to the periods with flow records upstream and downstream of the diversion dams. This is the case of the SJ River with flow records in SJ-km 47.3 and SJ-LTU located upstream and downstream of Dique la Roza (DLR) and Dique San Emiliano (DSE) diversion dams respectively for the period 1937-1951. Since the period under analysis was characterized by a significant flood in 1941/42 that contrasted with the low flows observed before and after (Figure 6), the HRI was determined for the entire period (1937-1951), for the period with high flows 1941-1946, and for the periods with low flows 1937-1940 and 1946-1951 (Table 7 and Figure 7).

In the CO river, the analysis was applied by comparing the average monthly flows in BRQ with those registered in PMA gauging station located downstream the Dique Punto Unido (DPU) diversion dam for the 1972-1990 period. PMA is located 550 and 360 km downstream of BRQ and DPU respectively. Flows showed a low magnitude attenuation downstream that




resulted in a MIF=0.879. The timing of maximum flows did not change (TIF=0.5) and the loss of interannual variability was
very low (VIF=0.464). These impact factors resulted in a HRI=0.84 that indicates a low impact on the hydrological regime.

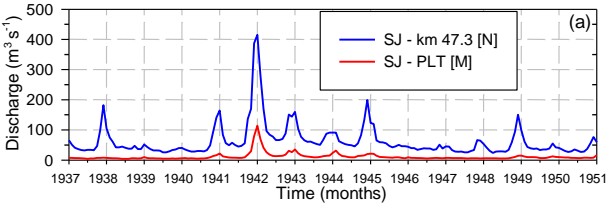
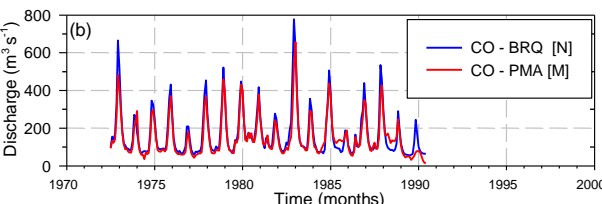

**Figure 6: Chronological monthly flows in natural [N] and modified [M] regime used to calculate the Hydrological Regime Index**
**(HRI) with low impoundment conditions in the tributaries of the Desaguadero-Salado-Chadleuvú-Curacó (DSCC) River and in the**
**Colorado (CO) River a) San Juan (SJ) River at km 47.3 and PLT gauging stations upstream and downstream of Dique la Roza**
**(DLR) and Dique San Emiliano (DSE) diversion dams respectively, b) CO River at BRQ and PMA gauging stations upstream and**
**downstream of Dique Punto Unido (DPU) diversion dam respectively.**
For the complete period, the MIF=0.270, TIF=0.417 and VIF=0.489 resulted in a HRI=0.24 that indicates a high impact on
the hydrological regime downstream the SLR diversion dam. However, if the previous and post flood conditions that better
represent the average flow conditions, are evaluated, the attenuation of the flow magnitude is very large (MIF=0.057). No
differences in timing were observed (TIF=0.417), but they contrasted with the drastic loss of natural variability downstream
(i.e. increase unnatural variability), where very low flows and only present during the summer season, differed from the almost
null and zero flows registered in the reset of the year (VIF=0). These impact factors determined an HRI =0.02 that illustrates
a drastic impact condition. Finally, if only the period with the highest flows is analysed, MIF=0.371, TIF=0.417 and
VIF=0.449. It gives an HRI=0.32 that corresponds to an equally high impact condition to the hydrological regime.
**Table 7: Hydrological Regime Index (HRI) for modified flows with low impoundment conditions in a tributary of the Desaguadero-**
**Salado-Chadileuvú-Curacó (DSCC) River and in the Colorado (CO) River. Qma: mean annual flow. [N]: natural flow, [M]: modified**
**flow. [us]: upstream, [ds]: downstream. MIF: Magnitude Impact Factor, TIF: Timing Impact Factor, and VIF: Variation Impact**
**Factor. San Juan (SJ) River at km 47.3 and Paso las Tunitas (PLT) located [us] and [ds] of Dique de la Roza (DLR) and Dique San**
**Emiliano (DSA) diversion dams respectively. CO River at Buta Ranquil (BRQ) and Pichi Mahuida (PMA) located [us] and [ds] of**
**Dique Punto Unido (DPU) diversion dam respectively.**

| River | Series | Qma (m³ s⁻¹) [N, us] | Qma (m³ s⁻¹) [M, ds] | MIF | TIF | VIF | HRI | Impact class |
|---|---|---|---|---|---|---|---|---|
| SJ | 1937-51 | 62.2 (km 47,3) | 16.8 (PLT) | 0.270 | 0.417 | 0.489 | 0.24 | High |
| SJ | 1937-40, 1947-51 | 45.9 (km 47,3) | 2.6 (PLT) | 0.057 | 0.417 | 0 | 0.02 | Drastic |
| SJ | 1940-46 | 80.1 (km 47,3) | 29.7 (PLT) | 0.371 | 0.417 | 0.449 | 0.32 | High |
| CO | 1972-1990 | 165.1 (BRQ) | 145.2 (PMA) | 0,879 | 0,5 | 0,464 | 0,85 | Low |




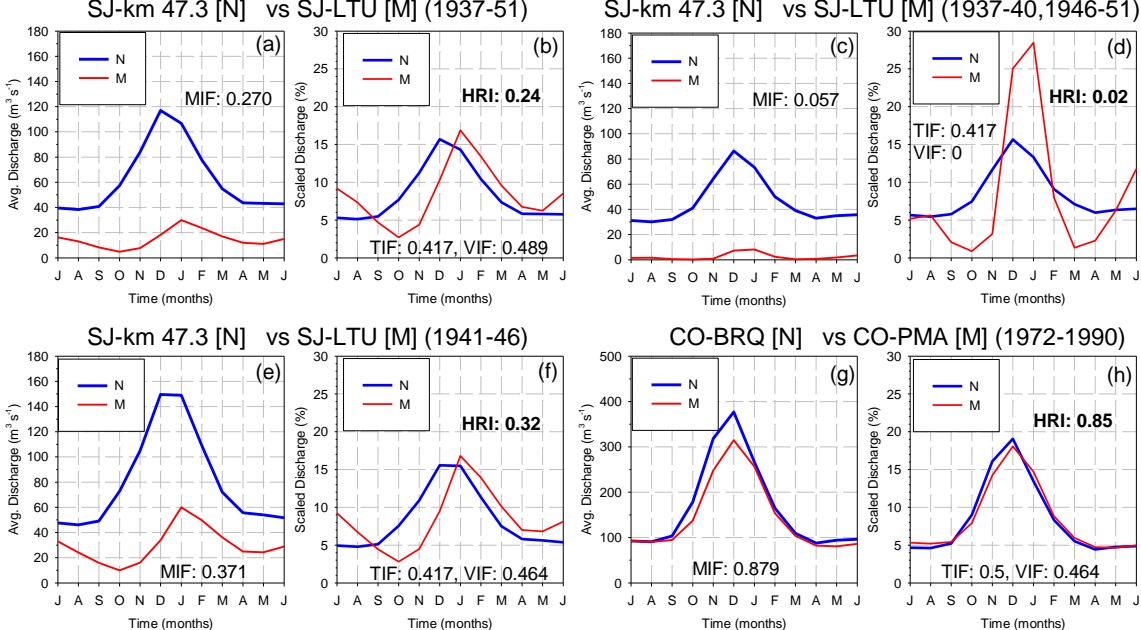

**Figure 7: Hydrological Regime Index (HRI) for low impoundment conditions. Annual and scaled hydrographs in natural [N] and modified [M] flows. MIF: Magnitude Impact Factor, TIF: Timing Impact Factor, and VIF: Variation Impact Factor. a-b) San Juan (SJ) River at km 47-3 [N] and PLT [M] complete period (1937-51), c-d) SJ River at km 47-3 [N] and PLT [M] low flow periods (1937-1940 and 1946-1951), e-f) SJ River at km 47-3 [N] and PLT [M] high flow period (1941-1946), and g-h) Colorado (CO) River at BRQ [N] and PMA [M] (1972-1990).**

**4.3 Hydrological regime index with high impoundment conditions**

The comparison of flow conditions upstream (i.e. natural regime) and downstream (i.e. modified regime) of the main reservoirs in the tributaries of the DSCC River and in the CO River revealed a different degree of modification of the hydrological regime (Figure 8). In general, in the tributaries of the DSCC there is no flow downstream the main reservoirs (e.g. JL, MZ, TY). In some of them, such us the SJ, the DT, and AT Rivers, intermittent runoff is observed downstream of the irrigation areas. However, this runoff is not natural but is the result of direct and diffuse drainage contributions from irrigation surpluses as a consequence of the use of very inefficient gravity irrigation systems. Therefore, flows show a strong attenuation or an intermittent condition with an inverted hydrological regime since they are mostly present in winter. This runoff disappears downstream and does not contribute to the DSCC River.

Furthermore, in the current period characterized by reduced natural flows, the effects described are more evident. The flows reduction showed a marked synchronicity in all the tributaries of the DSCC River and in the CO River, where consistent lower snowfall amounts in the CA were the result of the dominance of La Niña episodes.



Figure 8: Chronological monthly flows in natural [N] and modified [M] regime of the tributaries of the Desaguadero-Salado-Chadileuvú-Curacó (DSCC) River and in the Colorado (CO) River for available historical [H] and actual [A] (2010-23) periods. a) San Juan (SJ) River at km 47.3 [H, N], Mendoza (MZ) River at GUI [H, N] and SJ River at EEN [H, M], b) SJ at km 101 [A, N], MZ River at GUI [A, N] and SJ River at EEN [A, M], c) Diamante (DT) River at LJA [H, N] and MCO [H, M], d) DT River at LJA [A, N] and MCO [A, M], e) Atuel (AT) River at LAN [H, N] and PTU [H, M], e) AT River at LAN [A, N] and PTU [A, M], f) CO River at BRQ [H, N] and PMA[H, M], g) CO River at BRQ [A, N] and PMA completed with Casa de Piedra (CDP) reservoir discharges [A, M]

The HRI values were calculated based on the comparison between flows upstream and downstream of the main reservoirs for the historical and actual periods detailed in Table 8 and Fig. 9. For the SJ River, the natural flows in km 47.3 plus the contribution of the MZ River in GUI were contrasted with flows observed in the SJ River in EEN located downstream the of the QUL, DLR and DSE (in SJ River), POT, DSM and DCI (in MZ River) reservoirs and diversion dams, for the 1993-2010 period. In this condition, the mayor impact factor was the strong flow magnitude attenuation (MIF=0.174). In contrast, no changes in the maximum flow timing (TIF=0.5) and lover effects in the interannual variability were observed (VIF=0.449). The resulting HRI =0.15 indicates a severe impact on the hydrological regime. However, when actual conditions are analysed (2010-2023), the lower natural flows and the inclusion of the PTN and CAL reservoirs plus the construction ETA in the SJ



River, and the lack of contributions from the MZ River, exacerbated the effects downstream in EEN. Flows became intermittent
(MIF=0.014), with a strong effect in the timing given by the prevalence of winter flows (TIF=0) that resulted in a non-natural
variability (VIF=0). Consequently, the hydrological regime impact is classified as drastic with a HRI =0.
In the DT River, flows upstream the ADT, LRE, ETI, DGV, and DVI reservoirs and diversion dams, showed downstream in
MCO a high impact on the flow regime for the historical period (HRI=0.24) as a result of a MIF=0.369, TIF=0.417 and VIF=
0.225 values. For current conditions with no changes in the impoundment conditions, the lower natural flows resulted in a
stronger attenuation (MIF= 0.157), a marked delay on the occurrence of maximum flows (TIF=0.08) and a larger and non-
natural interannual variability due to the prevalence of winter flows (VIF= 0). The resulting HRI=0.16 indicates drastic effects
on the hydrological regime in MCO.
In the AT River, flows downstream the ENI, AIS, TBL, VGR and DRI reservoirs and diversion dams, showed for the historical
period a severe impact on its hydrological regime in CAR with HRI= 0.1. The marked attenuation (MIF=0.239) and the
dominance of winter flows (TIF= 0) were the main factors modifying the hydrological regime. For the current conditions, the
inclusion of the DCM diversion dam and the lower natural flows worsened the impact on the hydrological regime downstream
the reservoirs. The HRI degraded to a value of 0.07 indicating a drastic flow regime modification. In PTU, located 120 km
downstream of CAR, the HRI for the historical period equalled 0.01 indicated a drastic impact on the hydrological regime,
showing a strong flow attenuation, changes in timing and in the natural interannual variability (MIF=0.128, TIF=0 and VIF=
0.110). For actual conditions, the flow intermittence is more pronounced given by MIF=0.083, TIF=0 and VIF= 0 values,
which resulted in a HRI= 0 that indicates a maximum drastic impact.
The CO River showed for the historical period and incipient impact (HRI=0.62) on the hydrological regime of the flows in
PMA located downstream of the CDP and DPU reservoir and diversion dam. The flow attenuation resulted in a MIF= 0.791,
no changes were registered in the timing (TIF=0.5), however a marked reduction of the intraannual flow variability
(VIF=0.279) was observed presumably due to the filling of the CDP reservoir at the beginning of the period considered. In the
current condition with the same impoundment infrastructure, the lower natural flows resulted in a similar attenuation (MIF=
0.759), larger delay in maximum monthly values (TIF= 0.333) but a lower effect in the flow variability (VIF=0.423). The
resulting HRI equalled 0.57 indicating a moderate effect on the natural hydrological regimen n PMA.
**Table 8: Hydrological Regime Index (HRI) for modified flows in the tributaries of the Desaguadero-Salado-Chadileuvú-Curacó**
**(DSCC) River and the Colorado (CO) River with high impoundment conditions. Qma: mean annual flow. us: upstream, ds:**
**downstream. MIF: Magnitude Impact Factor, TIF: Timing Impact Factor, and VIF: Variation Impact Factor. San Juan (SJ) River**
**at km 47.3, km 101 and El Encón (EEN), Mendoza (MZ) River at Guido (GUI) and Cacheuta (CAC), Diamante (DT) River at La**
**Jaula (LJA) and Monte Común (MCO), Atuel (AT) River at La Angostura (LAN), Carmensa (CAR) and Puesto Ugalde (PTU), and**
**CO River at Buta Ranquil (BRQ), Pichi Mahuida (PMA) and Casa de Piedra (CDP).**

| River | Series | Qma (m³/s) [N, us] | Qma (m³/s) [M, ds] | MIF | TIF | VIF | HRI | Impact class |
|---|---|---|---|---|---|---|---|---|
| SJ+MZ | 1993-10 | 60.1 (km47.3) + 48.1 (GUI) | 18.8 (EEN) | 0.174 | 0,5 | 0,382 | 0,15 | Severe |
| | 2010-23 | 30.7 (km101) + 32.8 (GUI) | 0.9 (EEN) | 0.014 | 0 | 0 | 0 | Drastic |
| DT | 1990-10 | 31.2 (LJA) | 7.5 (MCO) | 0.369 | 0.417 | 0.225 | 0.24 | High |
| | 2010-23 | 19.1 (LJA) | 3.0 (MCO) | 0.157 | 0.080 | 0 | 0.01 | Drastic |
| AT | 1985-10 | 37.7 (LAN) | 9.0 (CAR) | 0.239 | 0 | 0.419 | 0.10 | Severe |
| | 2010-23 | 24.0 (LAN) | 3.9 (CAR) | 0.163 | 0 | 0.457 | 0.07 | Drastic |





| | | | | | | | | |
|---|---|---|---|---|---|---|---|---|
| AT | 1980-10 | 39.7 (LAN) | 5.1 (PTU) | 0.128 | 0 | 0.110 | 0.01 | Drastic |
| | 2010-23 | 24.0 (LAN) | 2.0 (PTU) | 0.083 | 0 | 0 | 0 | Drastic |
| CO | 1994-10 | 158.1 (BRG) | 125.0 (PMA) | 0.791 | 0.5 | 0.279 | 0.62 | Incipient |
| | 2010-23 | 79.1 (BRQ) | 59.3 (PMA/CDP) | 0.750 | 0.333 | 0.423 | 0.57 | Moderate |


**Figure 9: Hydrological Regime Index (HRI) of the tributaries of the Desaguadero-Salado-Chadileuvú-Curacó (DSCC) River and in the Colorado (CO) River with high impoundment conditions for available historical [H] and actual [A] (2010-23) periods. Annual and scaled hydrographs in natural [N] and modified [M] flows. MIF: Magnitude Impact Factor, TIF: Timing Impact Factor, and**

**VIF: Variation Impact Factor. a-b) San Juan (SJ) River at SJ-km 47.3 [H, N] plus Mendoza (MZ) River at MZ-GUI [H, N] vs SJ-EEN [H, M], c-d) SJ- km 101 [A, N] plus MZ-GUI [A, N] vs SJ-EEN [A, M], e-f) Diamante (DT) River at DT-LJA [H, N] vs DT-MCO [H, M], g-h) DT-LJA [A, N] vs DT-MCO [A, M], i-j) Atuel (AT) River at AT-LAN [H, N] vs AT-PTU [H, M], k-l) AT-LAN [A, N] versus AT-PTU [A, M], m-n) CO River at Buta Ranquil (BRQ) [H, N] vs Pichi Mahuida (PMA) [H, M], and o-p) CO River at Buta Ranquil (BRQ) [A, N] vs Pichi Mahuida (PMA) and Casa de Piedra [A, M].**

Although the DSCC River does not have large reservoirs, the severe flow regulation on its tributaries, determines que the DSCC River is dry. Flows are only presents as runoff pulses associated with ENSO episodes that eventually exceed the storage capacity of the reservoirs as occurred during the 1980s and particularly in 1988 and in less degree in 1998 and 2006 when the fluvial network of the DSCC River was fully activated. Fig. 10 depicts the longest time series available of monthly flows located in the lower part of the DSCC River basin. Based on these hydrological expressions, both the historical or reference period (1982-1992) with flows in natural regime and the actual period (2010-2023) characterized by their intermittent and very attenuated flow conditions, are indicated.

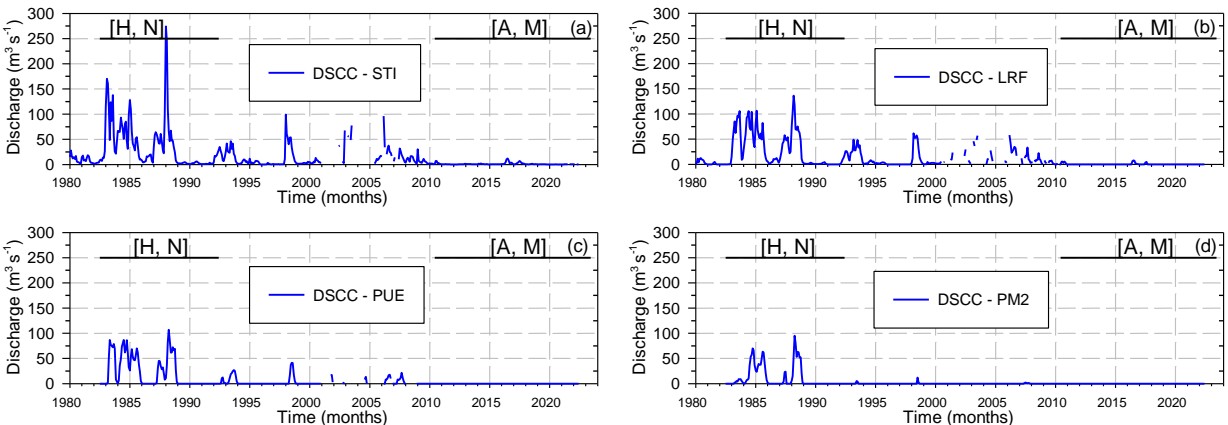

**Figure 10: Chronological monthly flows in the lower part of the Desaguadero-Salado-Chadileuvú-Curacó (DSCC) River basin with high impoundment conditions. Straight lines indicate the historical [H] period (1982-1992) in assumed natural [N] regime and the actual [A] period (2010-2023) with modified [M] regime. a) DSCC River at Santa Isabel (STI), b) DSCC River at La Reforma (LRF), c) DSCC River at Puelches (PUE), and d) DSCC River at Pichi Mahuida 2 (PM2).**

Table 9 and Fig. 11 compare both periods and indicate the values of the impact factors of the HRI in different gauging stations of the lower basin of the DSCC River. For the historical period, flows in STI showed a rather complex annual hydrograph with maximum flows in summer season. This is the result of the long travel time of allochthonous flows given by the extensive river network with respect to its headwaters (> 1000 km), and both anthropic (upstream reservoirs) and natural (LG and BT wetlands) flow regulation. In contrast, actual conditions have an almost uniform hydrograph showing a drastic flow attenuation (MIF= 0.032). The dominance of winter flows resulted in a TIF=0,084 whereas the intermittence of flows ended in a non-natural intrerannual variability that determined a VIF=0. The resulting HRI= 0.003 indicates a drastic impact determining a maximum flow regimen modification.

Downstream, the natural flows in LRF showed the combined effect of flow attenuation given by the regulation of BA wetland and the contribution of the AT River. This regulation resulted also in a complex hydrograph with summer and winter flows





slightly more uniform than the observed in STI. However, for the current conditions, intermittent conditions with much
attenuated winter flows were observed, The MIF= 0.013, TIF= 0.167 and VIF= 0 indicate the attenuation of the flows, their
intermittency and winter occurrence. These impact factors resulted in a HRI= 0.002 showing a drastic impact of the
hydrological regime.
As can be observed in both PUE and PM2 gauging stations, the annual hydrograph of the historical period showed highly
dominance of winter flows given by the natural flow regulation of the LP wetland. On the contrary, no flows were observed
for the actual conditions along the 13 years considered. In consequence, all the impact factors and the resulting HRI equalled
0, indicating a drastic impact on the hydrological regime in both gauging stations.
**Table 9: Hydrological Regime Index (HRI) for modified flows in the Desaguadero-Salado-Chadileuvú-Curacó (DSCC) River with**
**high impoundment contitions. Qma: mean annual flow. [N]: historical period (1982-92), [A]: actual period (2010-23). MIF:**
**Magnitude Impact Factor, TIF: Timing Impact Factor, and VIF: Variation Impact Factor. DSCC River at Santa Isabel (STI), La**
**Reforma (LRF), Puelches (PUE) and Pichi Mahuida 2 (PM2).**

| River | Qma ($m^3 s^{-1}$) [H] | Qma ($m^3 s^{-1}$) [A] | MIF | TIF | VIF | HRI | Impact Class |
|---|---|---|---|---|---|---|---|
| DSCC-STI | 37.5 | 1.2 | 0.032 | 0.084 | 0 | 0.003 | Drastic |
| DSCC-LRF | 30.2 | 0.4 | 0.013 | 0.167 | 0 | 0.002 | Drastic |
| DSCC-PUE | 22.2 | 0 | 0 | 0 | 0 | 0 | Drastic |
| DSCC-PM2 | 12.0 | 0 | 0 | 0 | 0 | 0 | Drastic |


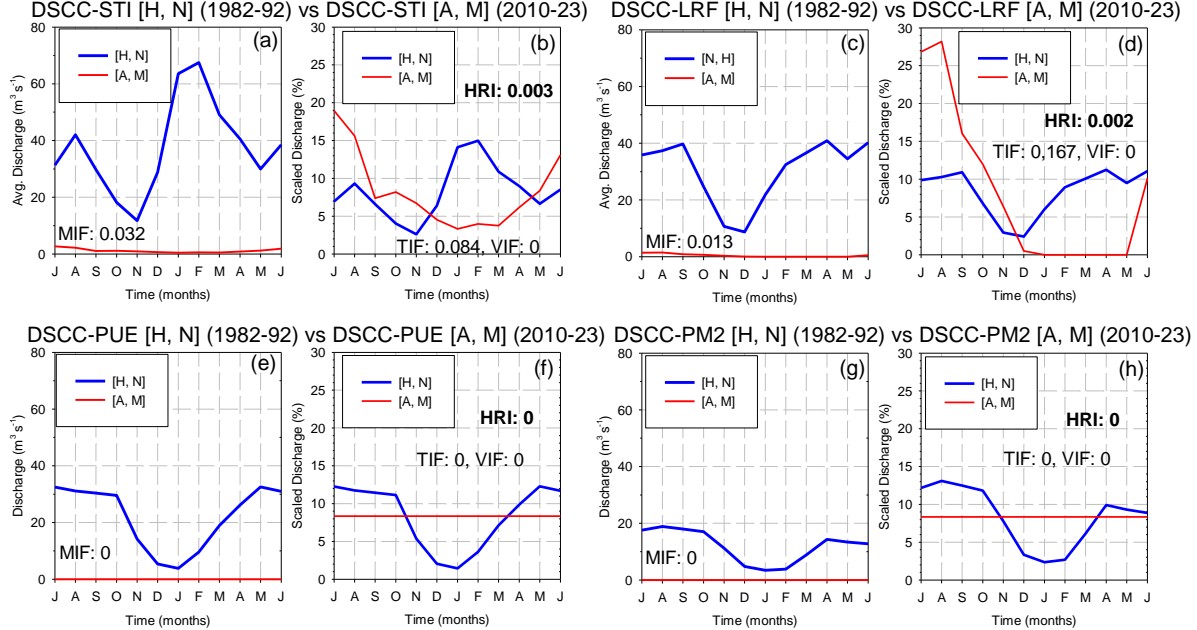

**Figure 11: Hydrological Regime Index (HRI) of the lower basin of the Desaguadero-Salado-Chadileuvú-Curacó (DSCC) River with**
**high impoundment conditions. Annual and scaled hydrographs for the historical [H] period (1982-92) with natural flows [N] and**
**actual [A] period (2010-23) with modified [M] flows. MIF: Magnitude Impact Factor, TIF: Timing Impact Factor, and VIF:**
**Variation Impact Factor. a-b) DSCC River at Santa Isabel (STI) [H, N] versus [A, M], c-d) DSCC River at La Reforma (LRF) [H,**
**N] versus [A, M], DSCC River at Puelches (PUE) [H, N] versus [A, M], and DSCC River at Pichi Mahuida 2 (PM2) [H, N] versus**
**[A, M].**





## 5. Discussion

Hydrological river regime is a spatial and temporally integrated basin response, therefore a comprehensive approach should be used to assess the impacts due to both anthropogenic interventions such as river regulation and water diversion for different uses and change in climate conditions.

As described, there are several point measurements of change in the flow regime. They are usually based on simple characteristics or statistics of the river flow hydrograph, such as mean, maximum and minimum flow values, CV, and flow frequency for a given percentage of time, whereas flow variation is usually addressed by establishing ratios between some of these parameters or by the average flow in a given season. These metrics do not necessarily represent the flow distribution over the hydrological or water year in different conditions. Indeed, under the intermittence of flow for long periods, some of the statistics are not appropriate (e.g. CV cannot be mathematically solved under non-flow conditions, the flow duration curve is a straight line of zero flow for the period considered). Similarly, the occurrence of unnatural variability (e.g. contributions of temporally lagged drainage from irrigation areas) may not necessarily be captured by seasonal averages.

The proposed HRI is a single and dimensionless metric that considers the impacts on the annual distribution of flows, which is by definition the hydrological regimen. Therefore, monthly mean flows are used to evaluate the different impact factors. This method allows its application in large basins, where daily flow variations do not necessarily represent the river-aquifer interaction, or the activation of a wetland, or the maintenance of ecosystem functions downstream the reservoirs. Additionally, this approach allows addressing the usual lack of daily flow data. It is a flexible method based on the comparison of sites or time series of the flow magnitude (i.e. attenuation of flow), the timing of maximum flow (i.e. occurrence of the peak flow) and annual variation of flows (i.e. temporal pattern of flow variability). Conceptually, HRI is similar to the index proposed by Haghighi Toraby and Kløve (2013) and Haghighi Toraby et al (2014), however, a simpler approach is used to compute the river regime index. HRI does not use conceptual hydrographs and somehow complex functions to represent the monthly river regime. Instead, the differences between the natural o reference regime and a uniform regime representing full regulation or no-flow conditions are calculated.

The HRI proved to be a suitable indicator due to its ability to discriminate both the spatial and temporal impacts on the hydrological regime in the DSCC and CO Rivers under continuous and discontinuous flow conditions and different degrees of regulation or impoundment conditions. For natural regime, the synchronous comparison of flows between upstream and downstream gauging stations in the tributaries of the DSCC River showed the sensibility of the HRI. The HRI values indicated incipient or extremely low impact on the hydrological regime as a result of minimal or no attenuation of flows, minimal time lag of the maximum flow and a reduced loss of interannual variability. The analysis could have been done based on the asynchronous comparison of flow series; however, this criterion was applied to consider equal hydrological conditions. In the case of the AT River, the incipient modification resulted from the streamflow losses in the alluvial fan at Las Juntas and the important distance (> 100 km) between the gauging stations. Beyond the applicability or the approximation used to determine




HRI for flows in a natural regime, it is verified that HRI can be a useful management tool to quantify impacts caused by
changes in land and water use.
This approach was validated in the CO River with higher flows. In its tributaries, the natural flow inputs of the GD and BR
Rivers, did not show a modification downstream in BRQ neither in the flow attenuation, nor in the timing or in the flow
variability. The resulting HRI indicated an impact extremely low on the hydrological regime. Similarly, on its main channel
between BRQ and PMA, the impact on the hydrological regime was classified as low, even though both gauging stations are
located 550 km apart and there was water allocation for consumption and irrigation.
The HRI applied to low impoundment conditions demonstrated its sensitivity to different hydrological conditions. When
extreme flows occurred, presumably associated with the ENSO episode, such as those in 1941/42, flows exceeded the capacity
of the reservoir and water diversion and the impact on the regime hydrological downstream was high. On the contrary, with
average natural flows, the impact observed downstream of the diversion dam was drastic. This showed that the operation of
these hydraulic structures played an important role as well, mainly due to the high seasonal demands given by the low
efficiency of gravity-fed irrigation systems.
For high impoundment conditions, the HRI adequately discriminated the reduced or no flow observed downstream of the
reservoirs in the DSCC River. The HRI values indicated severe and drastic impacts in all the tributaries. Moreover, the
different impacts were quantified and identified, such as the severe attenuation of the flow magnitude with an average MIF
value for all the tributaries of 0.228, and the inversion of the regime with winter flows and zero summer flows that resulted in
very low values or equal to zero of TIF and VIF. This indicates that the runoff observed immediately downstream of the
irrigation areas is not natural runoff but rather comes from irrigation drains. Towards downstream the modification of the
hydrological regime worsened and flow becomes more intermittent until runoff disappears.
For the current conditions characterized by lower natural flows, and considering that the reservoir conditions did not change,
except in the SJ River, a degradation of HRI values was observed in all tributaries. Although these values could be attributed
to climate change that resulted in lower flows, the HRI impact factors demonstrated that water management for irrigation is
the main cause of the drastic alteration of the snow-fed flow regimen observed in the tributaries. The MIF impact factor resulted
in values close to or lower than 50 % of the values obtained in the historical series, as a result of the reservoir operation for
irrigation purposes with a total diversion of water stored during the crop growing season and water storage in the rest of time.
The effects of hydropower management that could affect frequency and duration flow pulses and the rate and frequency of
change in the flow cannot be properly assessed because flow downstream of the irrigation areas is not natural but rather comes
from drainages or eventual releases of water. This resulted in values of TIF and VIF equal to 0 or between the historical and
current series. The exception is CAR with the same VIF values, which due to its location immediately downstream of the
irrigation area, already presented unnatural variability in the historical series. As indicated for historical periods, the incidence
in the modification of the hydrological regime is given by the high water demands of the gravity irrigation systems due to their
low efficiency. Thus, in years of lower natural contributions, the impact on the hydrological regime is more evident. In the





DSCC River, the lack of flow determined that the HRI values were equal to zero indicating a drastic impact in all the gauging
stations.
In the CO River, lower natural flows also resulted in a degradation of the HRI indicating a moderate impact in relation to the
incipient impact observed in the historical period. Changes in natural runoff also showed an advance in the occurrence of
maximum flows due to both a shorter extension of the snow accumulation period and a rapid ablation of snow. Therefore,
downstream of the CDP reservoir, similar values of attenuation and intra-annual variability were observed with a small increase
in the temporal lag of the maximum monthly flows.
From a simple visual inspection, it is obvious that there is a drastic modification of the hydrological regime in the DSCC River,
however, the HRI allows us to quantify the effects and discriminate the impacts on the natural hydrological regime. In this
way, it constitutes a very useful tool for defining E-Flows and for evaluating the efficiency of structural and non-structural
management measures that should be implemented to restore the environmental damage of the fluvial ecosystems of the DSCC
River caused by the absence of runoff.
Climate change is another critical factor of regime modification. Predictions for the study area indicate less snowfall and an
increase in rainfall. However, according to Arheimer et al (2017), the anthropogenic influence on the snow-fed hydrological
regime of the DSCC River proved to be severe with respect to the possible effects of climate change on the input function of
the basin. Therefore, for sustainable freshwater management, the proposed HRI will contribute to focus on the adaptation to
climate change and other environmental stressors (Poff and Matthews, 2013) such as the lack of integrated water resources
management in the basin.
**6. Conclusions**
An index, the HRI, is presented to evaluate the modifications of the hydrological regime in non-permanent rivers. The usually
drastic flow alterations in rivers of semiarid regions, where runoff can alter between a permanent or intermittent flow condition,
require a new approach to properly evaluate the modification of the hydrological regime in these basins which typically have
limited information. The HRI constitutes an aggregate impact index that allows its spatial or temporally distributed use. It can
be applied at different points along the drainage network and is based on the comparison of flow upstream and downstream of
the reservoirs, whereas the comparison between different time series makes it suitable to evaluate variations in impoundment
conditions, changes in land use or climate change effects.
The HRI evaluates three impacts on the hydrological regime: the attenuation of the flows, the time lag, and the change in the
intra-annual variability. It is based on the comparisons of monthly data which facilitates its application in areas with limited
information concerning other indices that use daily data.
The HRI was suitable for evaluating drastic flow alterations in the DSCC River under permanent and non-permanent flow
conditions in all the tributaries and in its main channel. Its application identified that, in addition to the impoundment
conditions, the operation of the reservoirs is one of the main modifying factors of the hydrological regime in the basin.





Additionally, the application of the HRI in the CO River under natural and modified flow conditions but with permanent
runoff, validated the method and showed the ability of the HRI to discriminate impacts between different hydrological
conditions.
The performance of the HRI in the DSCC river basin, characterized by the defined lack of hydrological connectivity between
the upper basin where the hydrological processes controlling the generation of snowmelt runoff in the mountain area are not
related to those in the lower basin where evaporative processes dominate, indicates that it is a valuable tool for E-Flow
definition and environmental impact assessment.
**Data availability**
All raw data is accessible in the digital databases indicated or it can be provided by the corresponding authors upon request.
**Author contributions**
PFD: conceptualization, data curation, formal analysis, investigation, methodology, validation, visualization, writing – original
draft preparation, writing – review and editing. RNC: data curation, visualization, review and editing.
**Competing interests**
The authors declare that they have no conflict of interest.
**Acknowledgements**
The authors wish to thank the Secretaría de Recursos Hídricos and the Secretaría de Ambiente de La Pampa, Argentina, for
providing flow data and information on the lower basin of the DSCC River.
**Financial support**
The research presented in this paper was conducted as part of the RN-50 project, funded by Facultad de Ciencias Exactas y
Naturales de la Universidad Nacional de La Pampa

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
