# Peer review of "Hydrological regime index for non-perennial rivers"

_Hydrology and Earth System Sciences, 2024_

## Author Comment (AC1)

**Final Author comments to RC2. HESS-2024-338**

**Major Comments**

-Comment 1: We understand the point about not explaining enough the existing 'gap' in the research and how the proposed metric (HRI) addresses it. As indicated by Reviewer 2 in RC2 the development of the HRI was motivated by the inability of existing metrics to quantify hydrologic regimes of non-permanent rivers. This is expressly indicated in the introduction section where it is described that the usual scarcity of detailed data (i.e. lack of distributed daily records, discontinuous flow record series, lack of records during no flow conditions) in semi-arid regions limit the use of flow alteration assessment indices (Leone et al., 2023; Gómez-Navarro et al, 2024).

Nonetheless, and to further guide the reader about the need to develop a new metric in intermittent rivers, the following paragraph was modified and added in the Introduction at the end of line 58:

"Indeed, indices based only on flow statistics, such as the interquartile variation range (IQR), the coefficient of variation (CV) or the flow duration curve (FDC), used as proxies for the seasonality of flows, among others; may not be suitable when no flow conditions are present. They require very detailed information not always available or not relevant to the dominant processes in the basin (e.g. allochthonous seasonal flow, interaction with groundwater), or they are based on complex theoretical functions of flow distribution of limited representativeness when runoff is not natural (e.g. only dams discharges; drainage flows), or the difficulty to standardize flows using statistical proxies (e.g. CV, IQR, FDC) for a given period when the average flow rate is zero. Therefore, new approaches to evaluate the modification of hydrological regimens in non-perennial rivers are needed. Two main reasons motivate this necessity. First and as indicated, the need to be able to mathematically solve relationships that adapt to intermittency flow conditions. Second, to have the capability to apply the index in a temporal and/or spatially distributed manner for the purpose of evaluating the hydrological connectivity in the large basins, which is a factor of fundamental importance for the quantification of E-flows."

Further, to expand the discussion of hydrological regime metrics in non-permanent rivers, in the Discussion section (Point 5) the second paragraph (lines 530-537) and third paragraph (lines 538-548) in page 24 were modified as indicated in the following:

"As described, there are several point measurements of change in the flow regime. They are usually based on simple characteristics or statistics of the river flow hydrograph, such as mean, maximum and minimum flow values, CV, and flow frequency for a given percentage of time, whereas flow variation is usually addressed by establishing ratios between some of these parameters or by the average flow in a given season. These metrics do not necessarily represent the flow distribution over the hydrological or water year in different conditions. Indeed, under the intermittence of flow for long periods, some of the statistics are not appropriate (e.g. CV cannot be mathematically solved under no flow conditions, the FDC curve is a straight line of zero flow for the period

considered). Similarly, the occurrence of unnatural variability (e.g. contributions of temporally lagged drainage from irrigation areas) may not necessarily be captured by seasonal averages or be assumed as natural if they are not compared with the flow upstream when a local or point evaluation index is used. Furthermore, the fact that many metrics are specific measurements limits the study of temporal and spatial hydrological variability, such as the analysis of hydrological connectivity that is usually lost in semi-arid basins under conditions of drastic flow alteration. This determines that the hydrological regime at a point downstream has no relationship with that upstream, a phenomenon that can hardly be evaluated with specific site measurements.

The proposed HRI is a single and dimensionless metric that considers the impacts on the annual distribution of flows, which is the more general definition the hydrological regimen. Therefore, monthly mean flows are used to evaluate the different impact factors. This method allows its application in large basins, where daily flow variations do not necessarily represent the river-aquifer interaction, or the activation of a wetland, or the maintenance of ecosystem functions downstream the reservoirs. Additionally, this approach allows addressing the usual lack of daily flow data especially during no flow conditions. Furthermore, it is not a specific measurement but is based on the comparison of flow records between upstream and downstream locations, a characteristic that allows evaluating, in addition to alterations resulting from hydraulic infrastructure, impacts of tributaries, interaction with groundwater, and the storage effects of wetlands. It is a flexible method based on the comparison of sites or time series of the flow magnitude (i.e. attenuation of flow), the timing of maximum flow (i.e. occurrence of the peak flow) and annual variation of flows (i.e. temporal pattern of flow variability). Conceptually, HRI is similar to the index proposed by Haghighi Toraby and Kløve (2013) and Haghighi Toraby et al (2014), however, a simpler approach is used to compute the river regime index. HRI does not use conceptual hydrographs and somehow complex functions to represent the monthly river regime. Instead, the differences between the natural o reference regime and a uniform regime representing full regulation or no flow conditions are calculated."

-Comment 2: Following the indication of the RC 2 about the manuscript organization, in the Introduction a new paragraph was included to guide readers regarding the expectation of being able to evaluate, through the HRI, the hydrological expression at different relevant points of the basin. Thus, the last paragraph of the Introduction (lines 77-82) was rewritten in two paragraphs as it is indicated in the following:

"Since the flow regime is an integrated basin response, a comprehensive approach should be used to evaluate its temporal and spatial distribution under both permanent and non-permanent flow conditions in areas with data scarcity. The hydrological metric must be able to describe the flow under natural (i.e. low modified) and modified conditions to varying degrees. For example, in the tributaries of the DSCC River the index must be capable to adequately discriminate between the hydrological conditions observed upstream and downstream of the main hydraulic structures. In the DSCC River, other hydrological characteristics also arise that must be appropriately evaluated, such

[revised manuscript text omitted]

-Comment 3: Regarding to the confusing terminology, we agree with the observations. Thus, 'actual' was changed to 'current' in the text, tables and figures. When possible, the ''natural' vs. 'modified' terms were replaced to 'upstream' vs. 'downstream'. However, it is not always de case, as most of the basin is downstream the main hydraulic structures. Besides, most of the times flow records upstream the reservoirs represent natural flows. Therefore, in the text natural flows (i.e. low modified flows) was used when necessary.

Furthermore, the terms low and high impoundment conditions are better described in section 3.2 (Data set) in the third paragraph (lines 285-298) as indicated above in the answer of Comment 2.

"The comparison between the flow records downstream of the reservoirs with those upstream in natural or low modified flow regime, was discriminated between periods with low impoundment conditions (i.e. storage reservoirs < 2 $hm^3$, see Table 2) where water for irrigation was derived mostly from diversion and small dams, and periods with high impoundment conditions (i.e. reservoirs with greater storage capacity, >100 $hm^3$) that represent current conditions (Table 5)."

**Minor Comments**

*Introduction*

-About the topic connectivity: we do consider that it is relevant in large semi-arid basins where the flow is usually intermittent. Therefore, to better describe this topic in Section 1 (Introduction) it is indicated that changes in the hydrological regime due to modification in basin connectivity was one of the motivations to develop the HRI. The inclusion is described in the above answer to Comment 1.

Moreover, in Section 2 (Study area) many implicit mentions to basin connectivity are already included, such us:

"…The DSCC River is distinguished by being an axial collector that receives on its right bank all its tributaries forehead mentioned and connecting important wetlands…." (lines 102 and 103): The functioning of the lower basin depends on the connectivity between the tributaries and the DSCC River itself.

"….The wetlands of the DSCC River are epigenic as a result of the fluvial contributions with null groundwater discharge…" (line 114): Therefore, its occurrence does not depend on local contributions but on river connectivity.

"The headwater of the DSCC River basin is the CA, where winter precipitation due to the orographic lifting of Pacific air masses by the mountains, constitutes the principal hydrological forcing of the basin". (lines 121 and 122). This paragraph was edited as will be described later in the following:

"This orographic configuration determines that the CA is the headwaters of the DSCC River basin, where winter precipitation due to the orographic lifting of Pacific air masses by the mountains, constitutes the principal hydrological forcing of the basin (Bruniard, 1986).": In both cases, it is implicit that fluvial connectivity is a key feature of the hydrological expression of the lower basin.

"….The runoff in the DSCC River is allochthonous due to the reduced rainfall that dominates the lower basin." (lines 162 and 163): The functioning of the lower basin depends on the connectivity between the tributaries and the DSCC River itself.

Moreover, in Section 3 (point 3.2 data set) and in the Discussion (point 5) the edition of the text included more explicitly this topic as it is described later in the Discussion section:

-Lines 27 and lines 29-31: We agree with the observations. Therefore, we edited the paragraphs trying to be more generalist and considering that large semi-arid basins are hardly fully activated (they usually do not depend only on climate configuration) but rather they typically present a more humid area where precipitation occurs (topographic configuration or proximity to the sea, etc., plus climatic variation) and the rest of the basin is dryer and usually has allochthonous flows. The modification is indicated below:

"In semi-arid regions, large basins are hardly fully activated since they usually do not depend solely on a climatic configuration. In contrast, there are other factors

such as relief or geographical location that determine the occurrence of precipitation. If the basin has a mountainous area, it usually constitutes the headwaters since precipitation is favoured by the orographic effect. Thus, the hydrological input function is restricted to those areas and almost none is manifested in the lower part. Moreover, higher temperatures result in important evapotranspiration losses which accentuate the hydrological deficit of the lower part of the basin."

-Lines 58-60: We modified the text for a better understanding. The term 'fail' is replaced in the following paragraph. Furthermore, a context for the description and examples of the limitations of the metrics in no flow conditions is provided in the above answer to Comment 1.

"Indeed, indices based only on flow statistics, such as the interquartile variation range (IQR), the coefficient of variation (CV) or the flow duration curve (FDC), used as proxies for the seasonality of flows, among others; may not be suitable when no flow conditions are present. They require very detailed information not always available or not relevant to the dominant processes in the basin (e.g. allochthonous seasonal flow, interaction with groundwater), or they are based on complex theoretical functions of flow distribution of limited representativeness when runoff is not natural (e.g. only dams discharges, drainage flows), or the difficulty to standardize flows using statistical proxies (e.g. CV, IQR, FDC) for a given period when the average flow rate is zero."

-Line 62: A location of the DSCC River basin is provided as indicated in the following:

"In this context, the Desaguadero Salado Chadileuvú Curacó (DSCC) River located in the central-west part of Argentina, provides a representative case study because it is an extensive semi-arid basin severely dammed which has undergone noticeable changes in its hydrological expression over the past century mainly due to the fragmented water governance along its transboundary water systems (Dornes et al., 2016)."

**Study Area**

As suggested, the section was reorganized considering the indicated contributions. It is an extensive basin that is characterized by both its relief and heterogeneous climate, which results in a complex hydrological expression and also with different levels of hydrological information. Therefore, we consider that it is important to provide an adequate description of these aspects to give the reader the possibility of understanding the results which are illustrated spatially distributed in the basin. Thus, first the aspects of the relief and climatic configuration were reorganized, then the description of the tributaries and finally the DSCC river itself and the wetlands. The headings of Table 2 and Figure 3 were also edited. The following paragraphs contains the reorganized Study Area Section:

[revised manuscript text omitted]

-Line 84: What is meant by "fully developed"? Could you just say "Argentina" instead of "Argentine territory" to improve clarity? The DSCC River basin is the largest basin entirely developed in Argentina. Other basins, such as La Plata River basin or even the Paraná River basin have greater areas in Argentina, but their total area also includes neighboring countries. Therefore, the following text seem to be appropriated:

"The DSCC River basin is the largest basin that extends entirely in Argentina"

-Line 90: Is the Cordillera de los Andes the same as the Andes mountain range (line 86)? The following text more appropriately describes what is indicated:

"The DSCC River basin is located in the central-west part of Argentina lying to the east of the Cordillera de los Andes (CA) mountain range with a north–south orientation (27° 47′ S, 38° 50′ S)….."

"…The DSCC River is located in the CA piedmont is defined by mountain ranges such as the Cordillera Principal, the Cordillera Frontal and the Precordillera to the West and North, the Sierras Orientales and Sierras Pampeanas to the East, whereas the lower basin is developed on flat terrain as part of the occidental area of the Pampean region (Ramos, 1999).

-Line 96: The precipitation doesn't contribute to the hydrology? Can that be true? The opposite is said in line 122.

The sentence was rewritten as indicated in the following:

"This orographic configuration determines that the CA is the headwaters of the DSCC River basin, where winter precipitation due to the orographic lifting of Pacific air masses by the mountains, constitutes the principal hydrological forcing of the basin (Bruniard, 1986). The rest of the basin is isolated from the influences of wet air masses driven by the extratropical high-pressure systems of the Atlantic and Pacific Oceans, a condition that results in an arid climate to the

North and semiarid to the South (Prohaska, 1976). These conditions generate a north-south precipitation gradient that ranges from values around 100 to 350 mm per year respectively, however this precipitation does not contribute to the average hydrological expression of the lower basin of the DSCC River which is strongly defined by the allochthonous snowmelt runoff from de CA (Dornes et al., 2016)."

-Line 100. "and the MZ River through the last one". Unclear what this means. The sentence was rewritten as indicated in the following:

"The DSCC River initiates as the outlet of the Lagunas de Guanacache (LG) wetland, which is fed by the VB, SJ and MZ Rivers (see Figure 1)."

**Materials and methods**

-Line 195: "…which is by definition the hydrologic regime". The modified text of the definition of the hydrological regime is described below:

"It is based on the comparison of the annual distribution of monthly flow records in natural or low modified with modified regimes (i.e. upstream vs downstream of a reservoir) which represent the long-term pattern of water flow and therefore the hydrological regime."

-Line 226: The modification of the description of the number of month (TQmN.max and TQmM.max) in equations 4 and 5 is detailed in the following:

"where TQmN.max and TQmM.max are the time (i.e. month number within the hydrological year) of occurrence of the monthly natural and modified maximum flow respectively".

-Line 230: Could you add a sentence somewhere to clarify what 0.0833 means in this equation? Why 0.0833 and not a different number?

The number 0.0833 is the slope of the linear relationship between the minimum value (TIF=0 when TD=0) and the maximum value (TIF=0.5 when TD=6). The following text was modified to clarify:

"To scale the TIF to a maximum value of 0.5 (i.e. natural flow) and a minimum value of 0 (i.e. maximum TD) applying a linear relationship with a slope of 0.0833 is calculated as following Eq. (6):"

-Line 239: Is Qsm always going to be equal to or greater than 8.33? If so, is it necessary to take the absolute value in eq. 8, since it should always be positive?

In the scaled hydrograph (Eq.1), Qms= 8.333 when the monthly quantities are equal or constant throughout the year. When the Qms vary throughout the year, its value may be greater or less than 8,333. Therefore, to analyse the interannual variability, the differences with respect to a constant flow are calculated. Consequently, the use of the difference in absolute value $MRI=|Qsm-8.333|$ ensures this calculation.

-Table 4: CAA is mentioned as an upstream gage in the table, but not in the text (Line 275); Do we need this table in addition to table 1? Or could they be combined? I'm also wondering if it would be useful/possible to label the gages (highlight in a different color) that are used in the analysis in either Figure 1 or 2 – not necessary, just a thought

The following paragraph shows the addition of the CAA gauge station in the text. "Based on the above and the availability of information, the MZ River at GUI and CAC (1956-90) and AT River at ESO plus the contribution of the Salado (SL) River at CAA respect to the records downstream in LAN (1972-03), were evaluated"

Table 4 tries to synthesize the information and details the common data period between gauging stations used in the calculation of the HRI in natural regime. Table 1 indicates location and the available record periods (historical and current) for each gauge station in the basin. Table 4 helps to link the Materials and methods section with the results (e.g. Figure 4).

Gauging station labels Figures 1 and 2 were edited highlining its colour (white in Fig 1 and black in Fig. 2)

- Paragraph at line 285: This is a difficult paragraph to follow. Edit to improve clarity.

The paragraph was edited to improve clarity as it is shown in the following. It is also showed in the above answer to Comment 2 where more context is provided.

"Further, to analyse the HRI performance in evaluating the impact of reservoirs on flow conditions, the HRI was applied in the DSCC River basin in two sectors, the tributaries and the lower reaches of the DSCC River, based on flow data availability. The effect of the reservoirs and their operation on the hydrological regime was contemplated for different impoundment conditions. The comparison between the flow records downstream of the reservoirs with those upstream in natural or low modified flow regime, was discriminated between periods with low impoundment conditions (i.e. storage reservoirs < 2 hm$^3$, see Table 2) where water for irrigation was derived mostly from diversion and small dams, and periods with high impoundment conditions (i.e. reservoirs with greater storage capacity, >100 hm$^3$) that represent current conditions (Table 5). In this case, only in the SJ River (km 47.3 vs PLT) was possible to evaluate the effect of a low impoundment condition from 1937 to 1950 and in the CO river (BRQ vs PMA) for the 1940-1971 period."

**Results**

-Paragraph at line 414, and elsewhere: There is a lot of explaining and providing contextual information in the results section…

The text that provided contextual information was edited so that the it refers to the results of the figures or tables. For example, the following paragraph shows the changes:

"The comparison of flow conditions upstream (i.e. natural regime) and downstream (i.e. modified regime) of the main reservoirs in the tributaries of the DSCC River and in the CO River revealed a different degree of modification of the hydrological regime (Figure 8). In tributaries, downstream of reservoirs and adjacent irrigation areas, runoff is intermittent. However, this runoff is not natural but is the result of direct and diffuse drainage contributions from irrigation surpluses as a consequence of the use of very inefficient gravity irrigation systems. Therefore, flows show a strong attenuation or an intermittent condition with an inverted hydrological regime since they are mostly present in winter. This runoff disappears downstream and does not contribute to the DSCC River."

**Discussion**

Line 549: Here and elsewhere, the authors claim that HRI is able to quantify the spatial impacts on the flow regime, but based on my reading, this is not true. No spatial analysis is incorporated into this metric. Please edit to clarify.

The determination of the impact factors (MIF, TIF and VIF) of the HRI are determined based on the comparison between flows upstream and downstream of a given point, which is why the HRI does consider spatial distribution. Likewise, the following paragraphs of the discussion were edited for clarity:

"The proposed HRI is a single and dimensionless metric that considers the impacts on the annual distribution of flows, which is the more general definition of the hydrological regimen. Therefore, monthly mean flows are used to evaluate the different impact factors. This method allows its application in large basins, where daily flow variations do not necessarily represent the river-aquifer interaction, or the activation of a wetland, or the maintenance of ecosystem functions downstream the reservoirs. Additionally, this approach allows addressing the usual lack of daily flow data especially during no flow conditions. Furthermore, it is not a specific measurement but is based on the comparison of flow records between upstream and downstream locations, a characteristic that allows evaluating, in addition to alterations resulting from hydraulic infrastructure, impacts of tributaries, interaction with groundwater, and the storage effects of wetlands……"

"….The HRI, due to its low data requirements and the determination of impact factors based on the difference between upstream and downstream flows, proved to be a suitable indicator to discriminate both the spatial and temporal impacts on the hydrological regime in the DSCC and CO Rivers under continuous and discontinuous flow conditions and different degrees of regulation or impoundment conditions"

-Line 600, elsewhere: The authors say that HRI is a useful tool for defining E-Flows, but it is not clear how HRI would be useful in this context. Please expand on exactly how HRI could be used to define e-flows.

The lack of runoff for long periods constitutes a drastic environmental impact on a basin and limits the application of hydrological metrics. One of the needs to avoid these impacts is the definition of E-Flows. In this context, the measures taken to mitigate these impacts must be able to be quantified to evaluate their effectiveness. This is where HRI is appropriate. The following paragraph indicates the changes made:

"From a simple visual inspection, it is obvious that there is a drastic modification of the hydrological regime in the DSCC River, however, the HRI allows us to quantify the degree of impact effects and discriminate the type of impacts (i.e. attenuation of flows, time lag of maximum flows and reduction of variability) on the natural hydrological regime. In this way, the determination of these parameters contributes to the definition of E-Flows. Therefore, the HRI constitutes a very useful tool for evaluating the efficiency of structural and non-structural management measures that should be implemented to restore the environmental damage of the fluvial ecosystems of the DSCC River caused by the absence of runoff."

-Line 603: This paragraph feels disconnected from the rest. Please consider either removing, or expanding to better integrate it.

Since climate change (i.e. modification of hydrological forcing) can be an impact factor on the hydrological regime, the current period of data was analyzed. However, the effects of anthropic regulation dominate the hydrological expression of the DSCC River, similar to what was expressed by Arheimer et al (2017). The following paragraph indicates the changes made:

"Climate change is another critical factor of regime modification whose effects can be evaluated with the HRI. The current period showed less runoff as a result of less snowfall in the basin and predictions for the study area indicate less snowfall and an increase in rainfall. However, according to Arheimer et al (2017), the anthropogenic influence on the snow-fed hydrological regime of the DSCC River proved to be severe with respect to the possible effects of climate change on the input function of the basin. Therefore, for sustainable freshwater management, the proposed HRI will contribute to focus on the adaptation to climate change and other environmental stressors (Poff and Matthews, 2013) such as the lack of integrated water resources management in the basin."

**Technical**

- Line 87: Change 'partial' to 'partially': Changes were applied.

- Line 101: Change "till" to "until": Changes were applied.

- Line 103: I believe you mean "aforementioned", not "forehead": Changes were applied.

- Line 198: Change "similarly" to "similar": Changes were applied.

- Line 203: typo: Changes were applied.

- You use different phrases when referring to no-flow events (no flow, non-flow). Consider standardizing the language used: Changes were applied, and no flow was used.

---

## Author Response (AR1)

Answer to Reviewer 1 comments

Overall, this research article presents a useful index to evaluate flow regime alteration. The paper is well written and well illustrated. I support the acceptance of the work as is.

We appreciate your contributions and take this opportunity to inform you that in accordance with the changes suggested by R2, as outlined below, along with the Editor's comments, the manuscript has been considerably enhanced.

Answers to Reviewer 2 Comments

**Major Comments**
In this study, the authors develop a single, comprehensive index to quantify the hydrologic regime in the DSCC River basin of Argentina. This metric, the Hydrological Regime Index, is useful in that it quantifies alterations in flow magnitude, timing, and variation. The development of this metric was motivated by the inability of existing metrics to quantify hydrologic regimes of non-permanent rivers.
While I think the authors make a fair argument that this type of metric would be useful, I don't believe that the authors communicate well enough what about their new metric makes it more useful than other metrics for non-permanent rivers. In other words, I think the authors need to better communicate the existing 'gap' in the research and how this metric addresses that gap. This is especially apparent in the discussion section, where there is minimal discussion of non-permanent rivers

We understand the point about not explaining enough the existing 'gap' in the research and how the proposed metric (HRI) addresses it. We expanded the description of why or where other metrics fail to accurately reflect alterations in the hydrological regime throughout the basin. As indicated by Reviewer 2 in RC2, the development of the HRI was motivated by the shortcomings of existing metrics in quantifying modifications of the hydrologic regimes in non-permanent rivers. This is expressly indicated in the introduction however it was not well developed in the discussion section. We enhance the interpretation of the results in this section showing the ability of the HRI to generate them under different hydrological conditions, particularly under no flow conditions. The English writing was improved throughout the paper.

Another weakness of the manuscript is the organization. The exception is the results, where the authors do a good job of walking through the results at the different types of gages where the metric was calculated and explaining why the findings make sense and are important. This manuscript could be improved if the authors had set up these expectations earlier, likely in the introduction. For example, a paragraph that explains what the authors expect a useful metric of hydrologic alteration to capture at each of the gages included in the analysis would help readers assess the results. Additionally, this would also allow the authors to refer to these expectations throughout the manuscript and eventually assess whether the Hydrological Regime Index met these expectations. I believe this would add a lot of much needed structure to the manuscript. As written, the methods section reads like the metric has been applied to many different gages for no clear reason. Making this change would help readers understand why each analysis is being done.

-The introduction was improved with a more comprehensive literature review and a better description of the limitations of existing metrics and capabilities of the HRI. The more detailed description of the basin as a case study where it is clearly indicated that

the flows are only present in the tributaries with no runoff in the lower basin, give to the readers a better interpretation of the results
-The study area as indicated was also reorganized as is detailed in the answer to minor comments and in track changes file.
-The methodology therefore was improved, mainly in in point 3.2 (data set) where a better description of the comparisons carried out between locations and over time was made. Also, it was described low and high impoundment conditions.
The results section included the changes indicated in the minor comments
-Discussion was strongly improved, linking the results with the expectations described in the Introduction. An additional figure was included as an illustration showing the capability of HRI in determining both the degree of and type of impact on the hydrological regimen even under long period with no flow.
-Conclusions: English writing corrections were effectuated.
-References. New references were included as a result of a more comprehensive literature review, and the specific references that were indicated in minor comments.

Finally, the terminology used throughout this manuscript tends to overcomplicate things, or even be slightly (unintentionally) misleading. Consider replacing the 'natural' vs. 'modified' language with 'upstream' vs. 'downstream'. In theory, you don't know if and how modified the flow downstream of a dam is until the analysis is completed. Additionally, flow upstream of dams is likely also modified by other factors. Also consider replacing what you call 'actual' with terminology like 'current' or 'present'. Even historical flow is 'actual', in that they are real, observed values. Using 'current' or 'present' sets up a better contrast with 'historical'. It's also never very clear to me what 'low impoundment' vs 'high impoundment' conditions means.

Regarding to the confusing terminology, we agree with the observations. Thus, 'actual' was changed to 'current' in the text, tables and figures of the manuscript. When possible, the ''natural' vs. 'modified' terms were replaced to 'upstream' vs. 'downstream'. However, it is not always the case, as most of the basin is downstream the main hydraulic structures. Besides, most of the times flow records upstream the reservoirs represent natural flows. Therefore, in the text natural flows (i.e. low modified flows) was used when necessary.
Furthermore, the terms low and high impoundment conditions are better described in section 3.2 (Data set) as indicated above. The impoundments conditions were also related to Table 1 to improve readers interpretation

**Minor Comments**
*Introduction*
A lot of this section discusses the topic of connectivity, but that's not relevant to the rest of this paper. Connectivity and hydrologic alteration are not the same, and the metric introduced in this paper does not measure connectivity. Consider removing most, if not all, mentions of connectivity.
About the topic connectivity: we do consider that it is relevant in large semi-arid basins where the flow is usually intermittent. Therefore, to better describe this topic in Section 1 (Introduction) it is indicated that changes in the hydrological regime due to the modification in basin connectivity was one of the motivations to develop the HRI. One of the reasons is that the DSCC River is distinguished by being an axial collector that receives on its right bank all its tributaries forehead mentioned and connecting important wetlands, which show an epigenic origin as they are the result of the fluvial contributions with null groundwater discharge. Therefore, hydrologic connectivity a key aspect. Efforts to integrate this feature in the manuscript, were set in the introduction, study area and discussion sections. In the track changes file, it is described in a more integrated manner.

Line 27: "In large basins, the headwaters of the drainage network are generally located in a mountainous sector that favours the occurrence of precipitation due to the orographic effect". Is this true? A lot of large basins don't contain mountains. Reference needed.

We agree with the observations. Therefore, we edited the paragraphs trying to adopt a more generalized approach and acknowledging that extensive semi-arid basins are hardly fully activated (they often do not rely solely on climatic setup) but instead usually feature a more humid region where precipitation takes place (topographical aspects or closeness to the ocean, etc., along with climatic fluctuations) while the remainder of the basin is drier and generally possesses allochthonous flows. The modification is indicated below:

"Arid and semi-arid basins typically present intermittent runoff in some sectors of the drainage network. This intermittence can be of different duration and extent (Datry et al., 2014; Boulton et al., 2017; Tramblay et al., 2021). Extensive semi-arid basins are hardly fully activated since they usually do not depend solely on a climatic configuration. In contrast, there exist other factors such as relief or geographical location that determine the occurrence of precipitation. Therefore, in large extensive complex terrain basins, the headwaters of the drainage network are generally located in a mountainous sector that favours the occurrence of precipitation due to the orographic effect. Consequently, the hydrological forcing of the basin typically occurs in the headwaters and almost none is manifested in the lower part (Viviroli and Weingartner, 2004)"

Lines 29- 31: Need to include references for these statements.
A reference (Viviroli and Weingartner, 2004) was included

Lines 58-60: "Indeed, indices based only on flow statistics…fail when no flow conditions are present". Reference needed. What is meant by 'fail'? These metrics can still be calculated when streamflow is equal to 0. This is the major gap you're working to address. I would like to see more evidence that this statement is true. Please expand more on how exactly other metrics fail and provide some examples.

We modified the text for a better understanding. The term 'fail' was replaced and examples were provided in the Introduction as it is indicated in the tack changes file.

Line 62: This is the first mention of the DSCC River. Consider adding some more information, like the country where it's located, to help orient the reader.

A location of the DSCC River basin is provided as indicated in the following:

"In this context, the Desaguadero Salado Chadileuvú Curacó (DSCC) River located in the central-west part of Argentina, provides a representative case study because it is an extensive semi-arid basin severely dammed which has undergone noticeable changes in its hydrological expression over the past century mainly due to the fragmented water governance along its transboundary water systems (Dornes et al., 2016)."

*Study Area*
There's a lot of information provided here in the text, tables, and figure. So much so that it becomes difficult to understand what is important because parts of this section aren't well organized. One example: the paragraphs that start at lines 137 and 161 both talk about El Nino, but the paragraph starting at 161 isn't really about El Nino, it seems. Consider finding ways to make the information provided in this

section more concise and more organized to help the reader understand what's important.

As suggested, the section was reorganized considering the indicated contributions. It is an extensive basin that is characterized by both its relief and heterogeneous climate, which results in a complex hydrological expression and also with different levels of hydrological information. Therefore, we consider that it is important to provide an adequate description of these aspects to give the reader the possibility of understanding the results which are spatially distributed in the basin. Thus, first the aspects of the relief and climatic configuration were reorganized, then the description of the tributaries and finally the DSCC river itself and the wetlands. The headings of Table 2 and Figure 3 were also edited as it is indicated in the document with the track changes.

Line 84: What is meant by "fully developed"? Could you just say "Argentina" instead of "Argentine territory" to improve clarity?

The DSCC River basin is the largest basin entirely developed in Argentina. Other basins, such as La Plata River basin or even the Paraná River basin have greater areas in Argentina, but their total area also includes neighboring countries. Therefore, the following text seem to be appropriated:
"The DSCC River basin is the largest basin that extends entirely in Argentina"

Line 90: Is the Cordillera de los Andes the same as the Andes mountain range (line 86)? If so, change so that you're using the same name to refer the range throughout.

The following paragraphs more appropriately describe what is indicated:
"The DSCC River basin is located in the central-west part of Argentina lying to the east of the Cordillera de los Andes (CA) mountain range with a north–south orientation (27° 47′ S, 38° 50′ S)….."

"…The DSCC River located in the piedmont of the CA is defined by mountain ranges such as the Cordillera Principal, the Cordillera Frontal and the Precordillera to the West and North, the Sierras Orientales and Sierras Pampeanas to the East, whereas the lower basin is developed on flat terrain as part of the occidental area of the Pampean region (Ramos, 1999).

Line 96: The precipitation doesn't contribute to the hydrology? Can that be true? The opposite is said in line 122.

The sentence was rewritten as indicated in the following:
"This orographic configuration determines that the CA is the headwaters of the DSCC River basin, where winter precipitation due to the orographic lifting of Pacific air masses by the mountains, constitutes the principal hydrological forcing of the basin (Bruniard, 1986). The rest of the basin is isolated from the influences of wet air masses driven by the extratropical high-pressure systems of the Atlantic and Pacific Oceans, a condition that results in an arid climate to the North and semiarid to the South (Prohaska, 1976). These conditions generate a north-south precipitation gradient that ranges from values around 100 to 350 mm per year respectively, however this precipitation does not contribute to the average hydrological expression of the lower basin of the DSCC River which is strongly defined by the allochthonous snowmelt runoff from de CA (Dornes et al., 2016)."

Line 100: "and the MZ River through the last one". Unclear what this means.

The sentence was rewritten as indicated in the following:
"The DSCC River initiates as the outlet of the Lagunas de Guanacahe (LG) wetland, which is fed by the VB, SJ and MZ Rivers (see Figure 1)."

Table 1: Why are there multiple mean annual discharge and record period values for some gauges? Historical vs. actual and modified vs. natural haven't been defined yet, so this table is really difficult to understand at this point. After a read through, I'm not convinced this table is necessary. The later tables about the gauges that are actually analyzed seem more useful.

Table 1 is necessary since it provides readers with the location of the gauging stations, the various flow magnitudes and the different recording period throughout the basin. This information is later utilized, and with the changes in the terminology of current and historic, natural and modified flows it can be interpreted more effectively. The following is the modification of Table 1 heading.
"Table 1: Mean annual discharge for the gauging stations (GS) in the Desaguadero-Salado-Chadileuvú-Curacó (DSCC) and Colorado (CO) Rivers. [H]: historical period, [C]: current period, [N]: natural flow (upstream the reservoirs), [M]: modified flow (downstream the reservoirs). VB: Vinchina-Bermejo River, JL: Jáchal River, SJ: San Juan River, MZ: Mendoza River, TY: Tunuyán River, AT: Atuel River, GD: Grande River, and BR: Barrancas River. VIN: Vinchina, PAC: Pachimoco, PLT: Paso las Tunitas, EEN: El Encón, GUI: Guido, VDU: Valle de Uco, LJA: La Jaula, MCO: Monte Comán, ESO: El Sosneado, CAA: Cañada Ancha, LAN: La Angostura, CAR: Carmensa, PTU: Puesto Ugalde, ADD: Arcos del Desaguadero, SLT: Salto de la Tosca, CAN: Canalejas, STI: Santa Isabel, LRF: La Reforma, PUE: Puelches, PM2: Pichi Mahuida 2, LGR: La Gotera, BAR; Barrancas, BRQ: Buta Ranquil, and PMA: Pichi Mahuida (PMA)."

Line 148: "…irrigation demands are unusually high". Reference needed.
The following paragraph incorporates the reference.
"The prevalent use of inefficient gravity-fed surface irrigation systems determines that irrigation demands are unusually high with respect to natural supply (Llop et al., 2013)."

Paragraph at line 161: It's not clear what this paragraph is about. Reorganize or incorporate the relevant information into other paragraphs.
This was done as part of the Study Area section editing as it was indicated before and in the track changes file.

Paragraph at line 179: You've said most of this information in other parts of this section. You also introduce the topic of salinization here, which isn't relevant to this manuscript. Consider cutting.
The sentence about salinization was eliminated

Figure 3: It still hasn't been said anywhere what defines historical vs. actual flows or natural vs. modified regime. Without that information, it's not clear what this figure is trying to communicate to the reader.
With the clarifications introduced in the Study Area section, the terms natural and modified flow can be adequately contextualized (see track changes file).
The following text is the Figure 3 foot.

"Figure 3: Annual hydrographs of the Desaguadero-Salado-Chadileuvú-Curacó (DSCC) River. a) Historical flows in natural regime of the tributaries of the DSCC, b) Current flows (2010-2023) in natural regime of the tributaries of the DSCC, c) Historical flows in modified regime of the DSCC, and d) Current flows (2010-2023) in modified regime of the DSCC River. Rivers, gauging stations, and historical and actual periods detailed in Table 1."

*Materials and methods*

Line 195: "…which is by definition the hydrologic regime". Not necessarily – you can characterize flow regime at finer or coarser temporal scales. Edit for clarity.
The modified text of the definition of the hydrological regime is described below:
"It is based on the comparison of the annual distribution of monthly flow records in natural or low modified with modified regimes (i.e. upstream vs downstream of a reservoir) which represent the long-term pattern of water flow and therefore the hydrological regime."

Line 226: definitions of TQmN.max and TQmM.max are unclear – what is meant by 'number of months'?
The modification of the description of the number of month (TQmN.max and TQmM.max) in equations 4 and 5 is detailed in the following:
"where TQmN.max and TQmM.max are the time (i.e. month number within the hydrological year) of occurrence of the monthly natural and modified maximum flow respectively".

Line 230: Could you add a sentence somewhere to clarify what 0.0833 means in this equation? Why 0.0833 and not a different number?
The number 0.0833 is the slope of the linear relationship between the minimum value (TIF=0 when TD=0) and the maximum value (TIF=0.5 when TD=6). The following text was modified to clarify:
"To scale the TIF to a maximum value of 0.5 (i.e. natural flow) and a minimum value of 0 (i.e. maximum TD) applying a linear relationship with a slope of 0.0833 is calculated as following Eq. (6):"

Line 239: Is Qsm always going to be equal to or greater than 8.33? If so, is it necessary to take the absolute value in eq. 8, since it should always be positive?
In the scaled hydrograph (Eq.1), Qms= 8.333 when the monthly quantities are equal or constant throughout the year. When the Qms vary throughout the year, its value may be greater or less than 8.333. Therefore, to analyse the interannual variability, the differences with respect to a constant flow are calculated. Consequently, the use of the difference in absolute value $MRI = |Qsm - 8.333|$ ensures this calculation.

Table 4: CAA is mentioned as an upstream gage in the table, but not in the text (Line 275); Do we need this table in addition to table 1? Or could they be combined? I'm also wondering if it would be useful/possible to label the gages (highlight in a different color) that are used in the analysis in either Figure 1 or 2 – not necessary, just a thought.
The following paragraph shows the addition of the CAA gauge station in the text.
"Based on the above and the availability of information, the MZ River at GUI and CAC (1956-90) and AT River at ESO plus the contribution of the Salado (SL) River at CAA respect to the records downstream in LAN (1972-03), were evaluated"
Although in the online response it was justified to maintain Table 4, subsequently and trying to be more concise, given that a figure (Fig. 12) was added to the manuscript in response to an editor's comment, Table 4 was eliminated considering that the information was already in Table 1.
Colour of tributary names in Figure 1 were changed. Since all the depicted gauging station (diamonds) in Figure 2 were used to compute the HRI, the following was added in the description of the figure.
"Diamonds: gauging stations used to compute the Hydrological Regime Index (HRI)".

Paragraph at line 285: This is a difficult paragraph to follow. Edit to improve clarity. The paragraph was edited to improve clarity as it is shown in the track changes file. It is also showed in the above answer to Comment 2 (mayor comments) where more context is provided.

*Results*

This section is well organized. If the methods followed the structure of this section, organization and readability would be improved.

Paragraph at line 414, and elsewhere: There is a lot of explaining and providing contextual information in the results section. I think this is relevant information, but I'm not sure the results section is the right place for it. Maybe this would fit better in the methods or in the discussion? The text that provided contextual information was edited so that it refers only to the results, to the figures or tables. In the document containing the track changes, modifications are indicated.

*Discussion*

Line 549: Here and elsewhere, the authors claim that HRI is able to quantify the spatial impacts on the flow regime, but based on my reading, this is not true. No spatial analysis is incorporated into this metric. Please edit to clarify. The determination of the impact factors (MIF, TIF and VIF) of the HRI are determined based on the comparison between flows upstream and downstream of a given point, which is why the HRI does consider spatial distribution. Along the section, a comparison is made between the abilities of metrics using point or local measurements and the HRI. As the section has been modified, it is recommended to review the changes in the track changes file or in the edited manuscript.

Line 600, elsewhere: The authors say that HRI is a useful tool for defining E-Flows, but it is not clear how HRI would be useful in this context. Please expand on exactly how HRI could be used to define e-flows. The lack of runoff for long periods constitutes a drastic environmental impact on a basin and limits the application of hydrological metrics. One of the needs to avoid these impacts is the definition of E-Flows. In this context, the measures taken to mitigate these impacts must be able to be quantified to evaluate their effectiveness. In the introduction it is described the necessity of E-Flows while the following text in Discussion section illustrates the point: "As detailed, the absence of runoff limits the utilization of hydrologic alteration metrics, as the majority of the parameters cannot be determined. For instance, the magnitude timing parameters remain unchanged due to all the average monthly flow are equal to zero, and this holds true for the magnitude duration (e.g. means of the annual maxima or minima). Likewise, this applies to magnitude frequency parameters such as the number of high or low pulses or their duration. Additionally, the parameters that assess the frequency rate of change (e.g. means of all positive or negative differences between consecutive daily means, or the number of rises or falls) remain unchanged. In this context, the HRI based on temporal or spatial monthly flow comparisons overcomes these limitations and therefore constitutes an essential tool for the

definition of E-Flows and for assessing the effectiveness of both structural and non-structural management measures that may be adopted to restore the environmental degradation of the fluvial ecosystems of the DSCC River caused by the absence of runoff."

Line 603: This paragraph feels disconnected from the rest. Please consider either removing, or expanding to better integrate it.
Since climate change (i.e. modification of the hydrological forcing) can be an impact factor on the hydrological regime, the current period of data (2010-2023) was analyzed. However, the effects of anthropic regulation dominate the hydrological expression of the DSCC River, similar to what was expressed by Arheimer et al (2017). The following paragraph indicates the changes made:
"Climate change is another critical factor of regime modification whose effects can be evaluated with the HRI. The current period has exhibited reduced runoff due to diminished snowfall in the basin, and predictions for the study area indicate a decrease in snowfall alongside an increase in rainfall. However, according to Arheimer et al (2017), the anthropogenic influence on the snow-fed hydrological regime of the DSCC River has been found to be detrimental in relation to the potential consequences of climate change on the input function of the basin. Therefore, for sustainable freshwater management, the proposed HRI will contribute to focus on the adaptation to climate change and other environmental stressors (Poff and Matthews, 2013) such as the lack of integrated water resources management in the basin."

**Technical**

- Line 87: Change 'partial' to 'partially': Changes were applied.

- Line 101: Change "till" to "until": Changes were applied.

- Line 103: I believe you mean "aforementioned", not "forehead": Changes were applied.

- Line 198: Change "similarly" to "similar": Changes were applied.

- Line 203: typo: Changes were applied.

- You use different phrases when referring to no-flow events (no flow, non-flow). Consider standardizing the language used: Changes were applied, and no flow was used

**Additional changes indicated by Editor Comments**

1) the novelty of your proposed method (HRI) should be highlighted, with more comprehensive literature review, and better introduction.

A more comprehensive literature review was conducted, focusing on the numerous existing indices of hydrological alteration (IHA) that primarily depend on various parameters whose statistics serve as indicators of flow alteration. Since these parameters may have intercorrelations, this could lead to statistical redundancy,

and potentially restricting their application. Additionally, in rivers located in semi-arid regions, obtaining such detailed information is very challenging, and many of the parameters of the IHA remain unchanged or cannot be mathematically resolved during no flow conditions. This is the research gap that requires further investigation.

In this way the Introduction section was improved to highlight the capacity of the proposed Hydrological Regime Index (HRI) in determining both the degree of and type of impact on the hydrological regimen even during extended periods of no flow. The low data requirements of the HRI, the evaluation based on the main impact factors of the hydrological regime (i.e. flow attenuation, time delay of maximum flows and interannual variability), and the formulation of the metric based on the spatial or temporal comparison of monthly flow series, allow for the identification of both the extent and type of impact on the hydrological regime. Additionally, a more detailed description of the basin as a case study where it is clearly indicated that the flows are only present in the tributaries with no runoff in the lower basin, give to the readers a better interpretation of the results

The document containing track changes, the answers to Reviewer 2 comments, and the final manuscript exhibit the modifications made to the Introduction Section as well as the remainder of the manuscript.

2) for the definition of non-perennial rivers, does it mean the streamflow is totally dry out, with zero streamflow? Or it means the volume of streamflow is under a threshold value?

The current conditions of the DSCC River are zero flow (i.e. streamflow is totally dry out) showing a tendency for increased intermittency in the last decade since almost no contributions of the tributaries were seen. This condition has been observed in the lower basin since 2006 as it is shown in Figure 10. In 2016, only the DT River contributed for 3-4 months to the DSCC River with a brief activation of the main channel on its central parts (DSCC-II and DSCC-III). To clear up doubts, the lack of flows was highlighted in the text in various parts of the document (Introduction, study area, results, and discussion). These conditions limit the use of many IHAs, and make the HRI, due to its ability to compare flow series in different hydrological conditions over time, capable of assessing both the degree and type of impact on the hydrological regime.

As it is indicated in the following, an additional figure (Fig. 12) has been incorporated into the manuscript based on remote sensing data that illustrates the extent of the river network with no flow, as well as the location of the main reservoirs. This inclusion was implemented following the indication of R2 that Table 4 was no longer necessary and thus was eliminated to maintain conciseness in the manuscript.

Do you think remote sensing data can benefit non-perennial river study to investigate its spatial distribution?

Yes, remote sensing data has the capability to illustrate the hydrological expression in extensive basins and thereby assist in contextualizing the modifications of the hydrological regime and its implications on the hydrological connectivity by providing information on rainfall, water levels, the extent of wetlands and lagoons, etc. Following this line and given that from a basic visual inspection of the DSCC

River basin, the total absence of runoff determines that there is a drastic impact, without the need to apply a given index, the use of remote sensing data can also show the capability of the HRI in evaluating the degree and type of impact on the hydrological regime. Indeed, Figure 12 was included to identify the varying water surfaces of the DSCC River network under contrasting hydrological conditions through the application of the Modified Normalized Difference Water Index (MNDWI) using optical satellite imagery. Moreover, remote sensing data could serve as an indicator of the impact factors of the HRI, like flow attenuation, timing of occurrence of maximum flow, or interannual variability, and it could also aid in monitoring ecohydrological processes, if representative relationships between the remote sensing products and impact factors are found. This was included in the discussion section.

3) the English writing should be thoroughly improved, to make the MS more readable and concise.

The English writing was enhanced. A complete revision of the manuscript was carried out to facilitate readers' comprehension. When necessary paragraph structure was adjusted or paraphrasing was performed.

---

## Author Response (AR2)

Answer to Editor comments

As the reviewer suggested, "Publish subject to technical corrections", "Presentation quality could be improved, especially conciseness."

**1. Introduction**
It was revised and a more standard paragraphing was applied for conciseness. Nevertheless, the text was mostly preserved in accordance with the reviewer's earlier comments to avoid confusion.

**2. Study Area**
A more concise paragraphing was used, especially in the text describing Figure 3.

**3. Materials and methods**
The text, mainly in point 3.2 (data sets), was revised to improve quality, especially the structure related to the results and conclusions, and for conciseness.

**4. Results**
The text was reorganized mainly in the analysis of the flows with low impoundment conditions section (Point 4.2).
Tables 7 and 8 headings were edited for clarity.

**5. Discussion**
The structure of the text was modified by unifying parts that were repeated, achieving greater clarity and a concise text, mainly in the justification of why HRI is adapted to non-permanent rivers compared to other hydrological metrics.

**6. Conclusions**
It was edited for clarity, and a more conventional paragraphing was used.